# The δ subunit and NTPase HelD institute a two-pronged mechanism for RNA polymerase recycling

Hao-Hong Pei [1], Tarek Hilal [2], Zhuo A. Chen[3], Yong-Heng Huang [1], Yuan Gao[1], Nelly Said [1], Bernhard Loll [1], Juri Rappsilber [3,4], Georgiy A. Belogurov [5], Irina Artsimovitch [6] & Markus C. Wahl [1,7✉]

Cellular RNA polymerases (RNAPs) can become trapped on DNA or RNA, threatening genome stability and limiting free enzyme pools, but how RNAP recycling into active states is achieved remains elusive. In *Bacillus subtilis*, the RNAP δ subunit and NTPase HelD have been implicated in RNAP recycling. We structurally analyzed *Bacillus subtilis* RNAP-δ-HelD complexes. HelD has two long arms: a Gre cleavage factor-like coiled-coil inserts deep into the RNAP secondary channel, dismantling the active site and displacing RNA, while a unique helical protrusion inserts into the main channel, prying the β and β′ subunits apart and, aided by δ, dislodging DNA. RNAP is recycled when, after releasing trapped nucleic acids, HelD dissociates from the enzyme in an ATP-dependent manner. HelD abundance during slow growth and a dimeric (RNAP-δ-HelD)$_2$ structure that resembles hibernating eukaryotic RNAP I suggest that HelD might also modulate active enzyme pools in response to cellular cues.

[1] Laboratory of Structural Biochemistry, Institute of Chemistry and Biochemistry, Freie Universität Berlin, Takustraβe 6, 14195 Berlin, Germany. [2] Institute of Chemistry and Biochemistry, Research Center of Electron Microscopy and Core Facility BioSupraMol, Freie Universität Berlin, Fabeckstr. 36a, 14195 Berlin, Germany. [3] Bioanalytics, Institute of Biotechnology, Technische Universität Berlin, Gustav-Meyer-Allee 25, 13355 Berlin, Germany. [4] University of Edinburgh, Wellcome Centre for Cell Biology, Edinburgh EH9 3BF, UK. [5] Department of Biochemistry, University of Turku, 20014 Turku, Finland. [6] Department of Microbiology and Center for RNA Biology, The Ohio State University, Columbus, OH, USA. [7] Helmholtz-Zentrum Berlin für Materialien und Energie, Macromolecular Crystallography, Albert-Einstein-Straße 15, 12489 Berlin, Germany. ✉email: markus.wahl@fu-berlin.de

Cellular RNA polymerases (RNAPs) are viewed as well-tuned engines that promptly re-initiate a new round of transcription after termination. For example, bacterial RNAPs minimally comprise an $\alpha_2\beta\beta'\omega$ subunit catalytic core, which forms a holoenzyme with one of several σ factors to initiate transcription at a promoter[1]. After promoter escape, elongation factors replace σ, and the ensuing elongation complex (EC) synthesizes RNA until a termination signal is reached. At a terminator, bacterial EC is abruptly destabilized either by an oligo-U-tailed G/C-rich RNA hairpin or by the RNA translocase/helicase ρ[2]. However, RNAP can linger on DNA after RNA release[3–5], road-blocking replisomes to trigger double-stranded DNA breaks[6] and giving rise to aberrant antisense transcripts[5]. RNAP can also form binary complexes with RNA[7,8], either through de novo association with stable RNAs, such as tRNAs and 6S RNA[9,10], or in the course of hairpin-induced termination[11]. While some RNA binary complexes serve as RNAP storage depots and can be reactivated when nutrients become available[10], others may sequester unproductive RNAP[12].

Post-termination binary complexes have to be dismantled to recycle RNAP, and ordered recycling is considered an integral phase of the duty cycle of many molecular machines, such as ribosomes[13]. By contrast, recycling has so far not garnered similar attention in bacterial transcription. While several accessory proteins could facilitate RNAP detachment from nucleic acids, including σ[8,9], transcription repair coupling factor Mfd[14], ρ[6], and the NTPase RapA[15], they release stalled RNAP under specific circumstances rather than act as genuine recycling factors.

RNAPs from some Gram-positive bacteria, including *Bacillus subtilis*, contain additional small nonessential subunits, δ and ε. δ is present in *B. subtilis* at an equal or higher concentration than standard core subunits, and its expression increases during the transition to the stationary phase[16,17], but δ deletion does not prevent sporulation[17,18]. Cells lacking the *rpoE* gene, encoding δ, have altered morphology and exhibit an extended lag phase[17] and defects in adaptation to changes in growth conditions sensed by initiating NTPs[19]. While a Δ*rpoE* strain has only mild phenotypes, it is not able to compete with the wild type (WT) strain[19], and δ is required for virulence in *Streptococci*[20,21]. δ destabilizes RNAP interactions with promoter DNA, inhibiting initiation at promoters that form unstable open complexes[19,22,23]. Consequently, δ suppresses initiation from weak or cryptic promoters, and deletion of *rpoE* leads to expression of many otherwise silenced genes in *Streptococci*[21,24]. Notably, δ also promotes RNAP recycling[22] by displacing σ from holoenzyme[25] and RNA or DNA from binary complexes[7]. Presently, it is unclear how δ elicits these effects. Likewise, the function of ε remains enigmatic[26].

HelD, a putative superfamily I nucleic acid-dependent NTPase found in Gram-positive bacteria, is related to *Escherichia coli* UvrD and Rep helicases[27] and has been implicated in DNA repair and recombination[28]. *B. subtilis* HelD and RNAP directly interact[29] and are present at comparable levels during sporulation[30]. Together with δ, HelD enhances RNAP cycling[29], and both proteins are required for adaption to environmental changes[19,29].

Based on the above, we hypothesized that HelD is a general recycling factor that acts in collaboration with δ and set out to elucidate its mechanism of action. Using single-particle cryogenic electron microscopy (cryoEM) and cross-linking/mass spectrometry (CLMS), we show that HelD, supported by δ, inserts long prongs into RNAP's main and secondary channels, competing with bound nucleic acids and prying RNAP open to allow nucleic acid escape. Release assays further support HelD/δ collaboration in RNAP recycling. ATP facilitates HelD detachment and completes RNAP recovery. We also observe RNAP dimerization in the presence of δ and HelD, hinting at a possible role of HelD in RNAP hibernation.

## Results

**Structural analysis of RNAP-δ-HelD complexes.** RNAP fractions enriched from stationary phase *B. subtilis* cells contained α, β, β', δ, ε, and ω subunits, with sub-stoichiometric amounts of HelD, PriA, $\sigma^A$ and $\sigma^B$ (Supplementary Fig. 1a). RNAP variants lacking HelD ($\text{RNAP}^{\Delta\text{HelD}}$) or lacking δ and HelD ($\text{RNAP}^{\Delta\delta\Delta\text{HelD}}$) were purified from *B. subtilis* Δ*helD* and Δ*helD*Δ*rpoE* strains, respectively (Supplementary Table 1); $\text{RNAP}^{\Delta\text{HelD}}$ contained δ and ε, yet showed a marked loss of ω (Supplementary Fig. 1b).

We assembled an RNAP-δ-HelD complex by supplementing stationary phase RNAP with δ, HelD, and a DNA/RNA scaffold with an artificial transcription bubble (Supplementary Table 1), followed by size exclusion chromatography (SEC). RNAP bound HelD but not the nucleic acid scaffold, and ω was again underrepresented in the RNAP-δ-HelD fractions (Supplementary Fig. 1c). CryoEM data were collected after vitrifying purified complexes without crosslinking in the presence of detergent to overcome preferred particle orientations (Supplementary Fig. 2). We iteratively extracted ~1,000,000 particle images from ~9100 micrographs for multi-particle 3D refinement (Supplementary Fig. 3a). Refinement led to two maps for monomeric RNAP-δ-HelD and dimeric (RNAP-δ-HelD)$_2$ complexes at global resolutions of 4.2 and 3.9 Å, respectively, with local resolutions extending beyond these limits (Supplementary Fig. 3b, Supplementary Table 2).

In both monomeric and dimeric complexes, we observed well-defined density for RNAP subunits α1/2 (N-terminal domains [NTDs]), β, β', δ, ε, and HelD (Supplementary Fig. 4). Density for the ω subunit or nucleic acids was missing. Unless mentioned otherwise, the following descriptions refer to the monomeric complex.

**Organization of RNAP in an RNAP-δ-HelD complex.** In the RNAP-δ-HelD complex, RNAP adopts a conformation in which the main channel, where downstream DNA and the RNA:DNA hybrid are accommodated in an EC, is wide open, with a distance of ~52 Å between the β2 lobe (P242) and the β' clamp helices (N283), compared to ~18 Å between the corresponding elements in the *E. coli* EC[31] (Fig. 1a, b and Supplementary Table 3), and a concomitant widening of the RNA exit tunnel by more than 17 Å (β flap$^{R800}$ to β' lid$^{D245}$). The α1/2$^{NTD}$ dimer remains bound at the closed end of the open β/β' crab claw.

δ consists of a folded N-terminal domain (NTD; residues 1–90) and an intrinsically disordered acidic C-terminal region (CTR; residues 91–173) with a net −47 negative charge[7,23]. As noted previously[32], the first ~70 residues of δ$^{NTD}$ resemble the globular domain of σ$^{1.1}$ regions of group 1 σ factors[33]. However, unlike the σ$^{1.1}$ domain in an *E. coli* σ$^{34}$ holoenzyme[34], δ$^{NTD}$ does not reside in the main channel but binds on the surface of RNAP between the β' shelf and jaw (Fig. 1a and Supplementary Fig. 5a), in agreement with a previous in vivo CLMS analysis[35]. Comparison to the *E. coli* EC[31] showed that δ$^{NTD}$ seems to contribute to main channel opening by somewhat contracting the jaw and β' shelf; furthermore, RNAP opening and slight δ$^{NTD}$-mediated displacement of the shelf lead to the repositioning of β' secondary channel elements, which would clash with ω at its canonical binding site (Supplementary Fig. 5a), explaining loss of ω in RNAP-δ and RNAP-δ-HelD complexes (Supplementary Fig. 1b, c). Lack of continuous cryoEM density beyond δ$^{NTD}$ shows that δ$^{CTR}$ is suspended from the rim of the main channel in a flexible manner (see below).

The ε subunit is positioned in a cavity formed by the α1/2 NTDs, the C-terminal β clamp, and β' residues 492–655 that form part of the secondary channel (Fig. 1a), in contrast to the previous mapping of ε at the β' jaw based on a low-resolution cryoEM

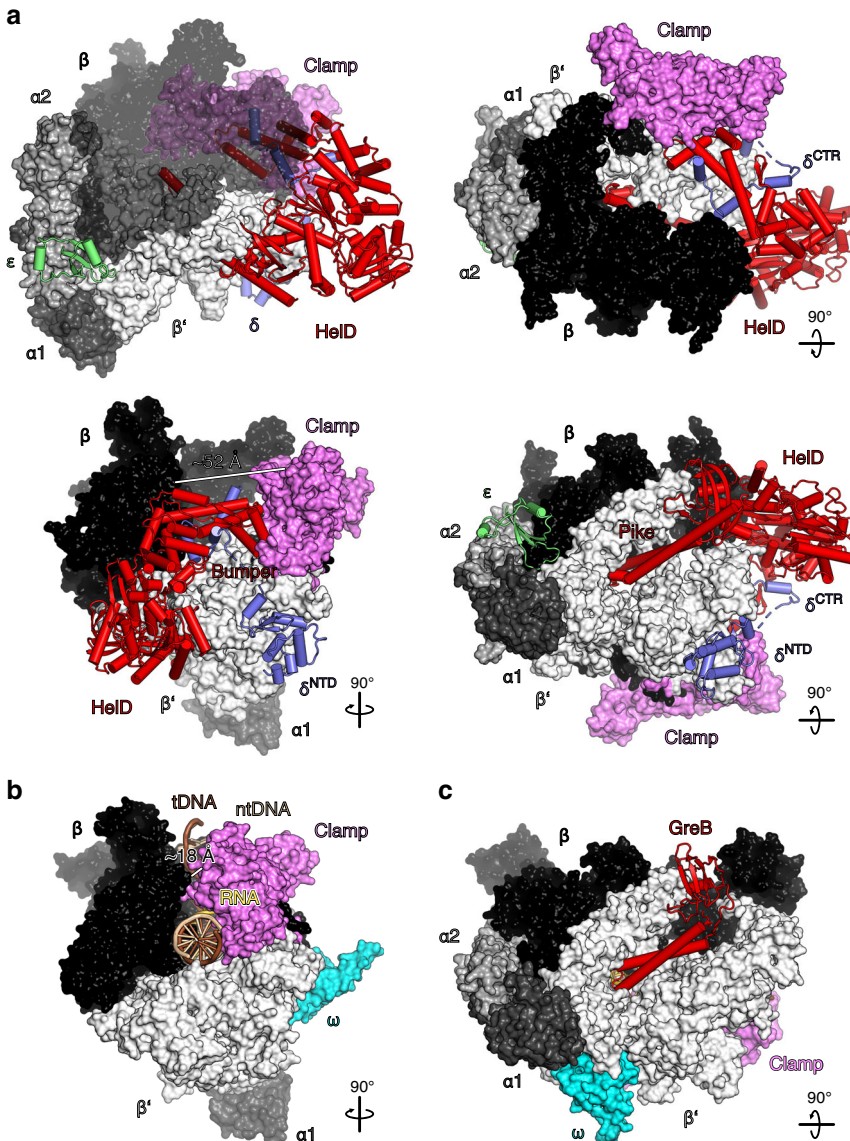

**Fig. 1 Structural overview. a** Overall architecture of an RNAP-δ-HelD complex. β surface is semi-transparent in the upper left panel. Rotation symbols in this and all figures indicate views relative to the upper left panel. Color coding in all figures, unless otherwise noted: α1, dark gray; α2, gray; β, black; β′, light gray; β′ clamp, violet; ε, lime green; δ, slate blue; HelD, red. **b** Comparison to an *E. coli* EC (PDB ID 6ALH), illustrating marked widening of the main channel in RNAP-δ-HelD. ω, cyan; template (t) DNA, brown; non-template (nt) DNA, beige; RNA, gold. **c** Comparison to an *E. coli* GreB-modified EC (PDB ID 6RIN), illustrating similar secondary channel invasion by coiled-coil elements in GreB and HelD.

analysis and structural similarity of ε to the phage T7 Gp2[26]. Interestingly, in archaeal and eukaryotic nuclear RNAPs this position is occupied by a domain of α1 subunit homologs (Supplementary Fig. 5b). We surmise that ε supports the structural integrity of RNAP, securing interactions between α, β, and β′ subunits when β and β′ are forced apart by HelD.

**HelD invades RNAP channels.** HelD consists of four domains/ regions: an N-terminal region (NTR; residues 4–187), two glob-ular domains (D1a/D1b, residues 188–338/491–603; D2, residues 604–774), and an elongated helical protrusion in D1 (HelD^Bumper; residues 339–490; Fig. 2a). The NTR exhibits remarkable resem-blance to GreA/B transcript cleavage factors, but with an extended coiled-coil (HelD^Pike; residues 4-96; Figs. 1c and 2b). D1 and D2 resemble NTPase/helicase domains of UvrD[36], with a subdomain deleted from D2 and HelD^Bumper inserted into D1 (Fig. 2c).

HelD^Bumper lacks close structural similarity to other proteins in the Protein Data Bank (https://www.rcsb.org).

HelD is reminiscent of a two-pronged fork poking into RNAP. In perfect analogy to transcript cleavage factors[37], one prong, HelD^Pike, inserts deeply into the secondary channel, through which substrate NTPs enter the RNAP active site during elongation (Fig. 1a, c). D1/D2 reach around the β2 lobe, positioning the other prong, HelD^Bumper, in the main channel where it pushes against the β′ clamp, forcing β and β′ apart (Fig. 1a). In the course of HelD engaging RNAP, a large combined surface area (~11,500 Å² total; ~8000 Å² with β′; ~1800 Å² with β; ~1700 Å² with δ) is buried.

We observed some cryoEM density patches around HelD^Bumper that could only be interpreted as parts of δ^CTR (Fig. 1a). However, the poor quality of the local cryoEM density did not permit reliable modeling of the precise region of δ^CTR that bound at HelD^Bumper. We confirmed a direct HelD-δ interaction via δ^CTR

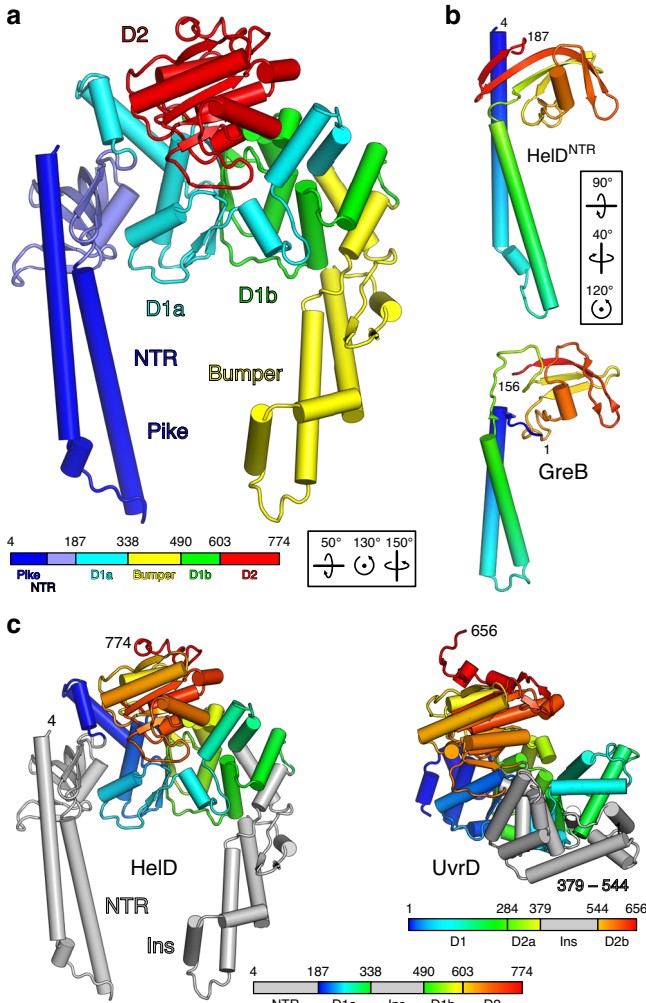

**Fig. 2 HelD architecture. a** Cartoon plot of HelD colored by domains (for color coding see legend). Numbers refer to domain borders. **b** Comparison of HelD[NTR] to GreB (PDB ID 6RIN) reveals similar topology of the coiled-coils, which insert into the secondary channel, and the globular domains; in GreB, the latter is responsible for high-affinity binding to the RNAP β′ rim helices. HelD[NTR] and GreB are rainbow-colored (blue, N-termini; red, C-termini). Numbers refer to domain borders. **c** Comparison of NTPase domains in HelD and in *E. coli* UvrD (PDB ID 2IS6). The D1-D2 regions are rainbow-colored (blue, N-termini; red, C-termini) as indicated in the legends. Neighboring and inserted regions (Ins), gray. Numbers refer to domain borders.

by analytical SEC; while HelD co-migrated with δ and the complex eluted earlier than the individual proteins (Fig. 3a), no such interaction was detected with δ[NTD] (Fig. 3b). These results suggest that δ[CTR] might help position HelD[Bumper] in the main channel, supporting HelD in its push against the β′ clamp (Fig. 1a). HelD[Bumper] and tentatively modeled portions of δ[CTR] reside in a position equivalent to the globular σ[1.1] domain in an *E. coli* σ[34] holoenzyme[34] and a helix following the σ[1.1] region in a *Mycobacterium smegmatis* σ[A] holoenzyme[38] (Fig. 3c–e). Thus, HelD[Bumper] and δ[CTR] occupy regions next to the β subunit where downstream DNA is accommodated in the EC (Fig. 3c).

Due to the combined actions of δ and HelD, RNAP-δ-HelD exhibits the most open main channel configuration observed in RNAP complexes to date, augmented by about 30 Å and 20 Å relative to the *E. coli* σ[34] and *M. smegmatis* σ[A] holoenzymes[34,38], respectively (Fig. 3c–e). To confirm contacts and the marked

structural rearrangements triggered by HelD binding, we used RNAP[ΔδΔHelD] and recombinant δ and HelD to assemble RNAP[ΔδΔHelD]-δ, RNAP[ΔδΔHelD]-HelD, and RNAP[ΔδΔHelD]-δ-HelD, and mapped molecular neighborhoods in these complexes and RNAP[ΔδΔHelD] by CLMS with the heterobifunctional, photoactivatable crosslinker sulfosuccinimidyl 4,4′-azipentanoate (sulfo-SDA; Fig. 4a, b and Supplementary Table 4; Supplementary Data 1). Matching the δ[NTD] binding site deduced by cryoEM, a short stretch of δ residues cross-linked to the β′ jaw in both RNAP[ΔδΔHelD]-δ (δ[Y82,P83,Y85]-β′[K1032]) and RNAP[ΔδΔHelD]-δ-HelD (δ[Y83,Y85,L87,E90]-β′[K1032]). Multiple crosslinks of HelD were identified for RNAP[ΔδΔHelD]-HelD and RNAP[ΔδΔHelD]-δ-HelD complexes inside the RNAP main channel, along the region connecting the main and secondary channels, and in the active site region, in excellent agreement with our cryoEM structures (Supplementary Fig. 6).

RNAP[ΔδΔHelD], RNAP[ΔδΔHelD]-δ, and RNAP[ΔδΔHelD]-HelD yielded significantly more crosslinks than RNAP[ΔδΔHelD]-δ-HelD and, among those, in particular, many more over-length cross-links when compared to the RNAP-δ-HelD structure (Fig. 4c, d). Furthermore, the fraction of crosslinks corresponding to over-length crosslinks was strongly increased in RNAP[ΔδΔHelD] and RNAP[ΔδΔHelD]-δ compared to complexes containing HelD (Fig. 4c, d). The reduced total number of crosslinks suggests a reduction in conformations explored by RNAP upon δ or HelD binding, and in particular when both factors are present. The reduced total number and fraction of over-length crosslinks suggest a conformation closer to our RNAP-δ-HelD cryoEM structure in the presence of HelD. A specific set of crosslinks between the β1/2 lobes (residues 146–248) and the β′ shelf and jaw (residues 794–1141) represents conformations in which β and β′ approach each other across the main channel unless both δ and HelD are bound to RNAP (Fig. 4e, f). Together, our results demonstrate that HelD interacts with the main and the secondary channels of RNAP and that stable main channel opening depends on the presence of both δ and HelD.

**HelD[Pike] dismantles the RNAP active site and competes with RNA.** Upon penetrating the secondary channel, HelD[Pike] locally disrupts the β′ bridge helix (BH; between residues 780 and 787) and locks the β′ trigger loop (TL; Fig. 5), i.e., key elements that rearrange for nucleotide addition during elongation[39]. While HelD[Pike] carries negatively charged side chains (D56, D57, E60) at its tip, these residues do not remodel the active site as observed with GreB[37]. Instead, the tip plows through the active site, thereby dismantling it. The β C-terminal clamp is pushed away from the nucleic acids, β switch region 3 (Sw3), which lines the hybrid in the EC, becomes disordered and the active site loop (ASL) is rearranged so that the catalytic Mg²⁺ ion is lost (Fig. 5).

RNAP-RNA binary complexes are catalytically active, implying that RNA resides in the active site cavity[8]. As seen by comparison with an *E. coli* EC[31], the HelD[Pike] tip binds in direct competition to RNA in the hybrid (Fig. 3f) and may additionally repel RNA via the negatively charged residues. Thus, HelD[Pike] rearranges active site regions and spatially competes with all RNAs bound in the vicinity. RNA release would be facilitated by RNA exit tunnel opening via HelD[Bumper].

**HelD[Bumper] and δ pry the main channel open and displace nucleic acids.** Clearly, the binding of HelD[Bumper] and δ[CTR] in the main channel is incompatible with DNA occupying this site (Fig. 3c). Previous studies had shown that δ or excess δ[CTR] alone can displace RNA or DNA from RNAP[7]. To further delineate the contributions of δ and HelD to nucleic acid displacement, we conducted band shift assays, in which we first bound RNAP to nucleic acids and

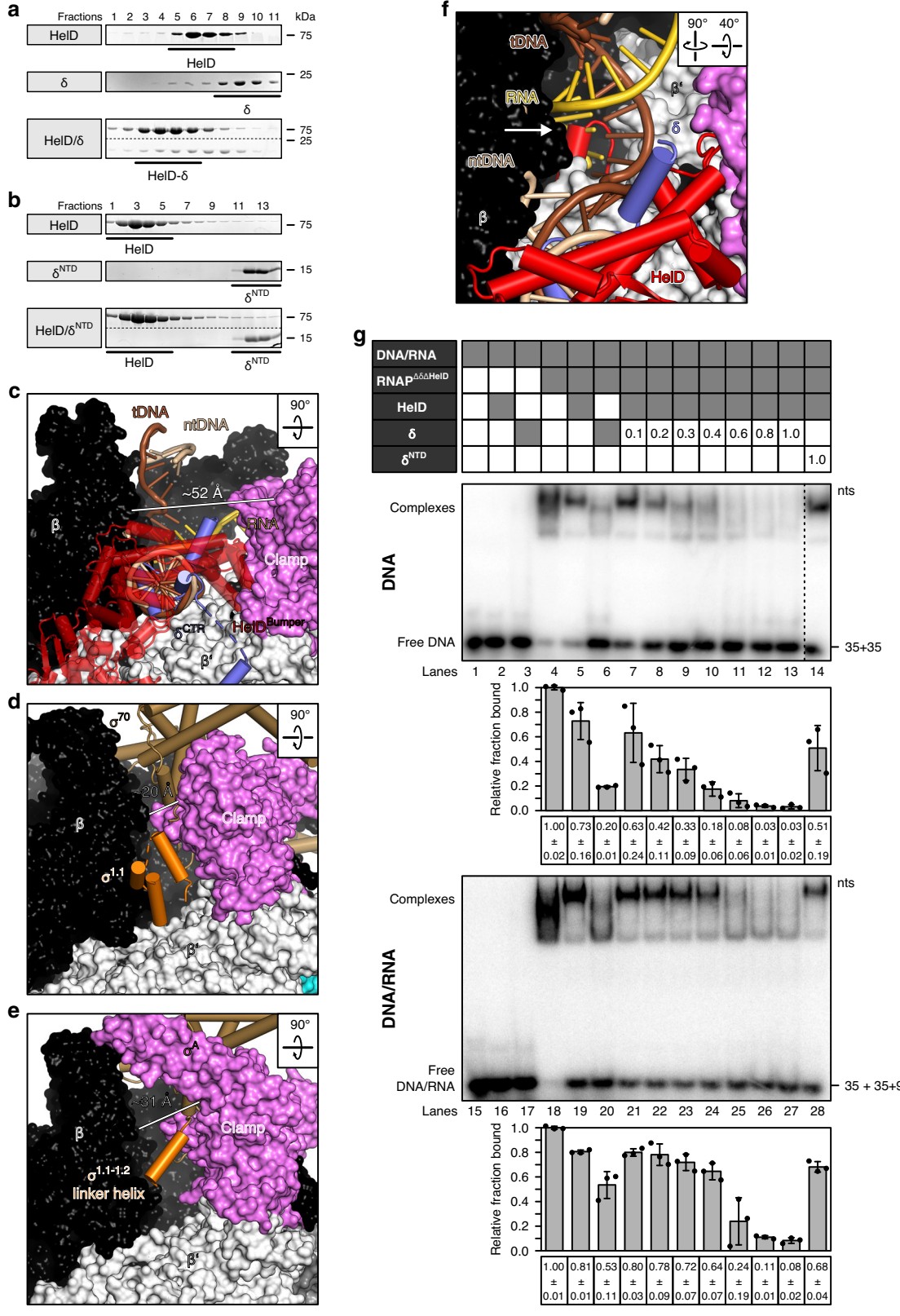

subsequently added δ and/or HelD. We first tested displacement of DNA with an artificial bubble, which when bound to RNAP mimics a situation ensuing after many intrinsic termination events[3–5,40]. HelD displaced about 25% of DNA from RNAP$^{\Delta\delta\Delta HelD}$, while δ led to about 80% displacement in the absence of HelD (Fig. 3g, lanes 4–6). Increasing amounts of δ titrated to DNA-bound RNAP$^{\Delta\delta\Delta HelD}$ in the presence of stoichiometric amounts of HelD led to a gradual

reduction of bound DNA, with essentially all DNA displaced when equimolar amounts of δ relative to RNAP$^{\Delta\delta\Delta HelD}$-HelD were added (Fig. 3g, lanes 7–13). Only ~50% of the DNA were displaced by the addition of equimolar amounts of HelD and δ$^{NTD}$ (Fig. 3g, lane 14).

Next, we tested the ability of δ/HelD to dissociate ECs assembled on an artificial DNA bubble and complementary RNA, mimicking stalled ECs. A similar picture as for DNA-only

**Fig. 3 HelD/δ-mediated RNAP recycling. a, b** SDS-PAGE monitoring SEC of a HelD/δ mixture (**a** lower panel), compared to a HelD/δ$^{NTD}$ mixture (**b** lower panel), compared to SEC runs of the isolated proteins (upper two panels). Analyzed fractions (numbers above the gels) were identical for the groups of three runs but different fractions were analyzed in **a**, **b**. In this and the following figures: kDa, molecular weight marker in kDa. **c** Nucleic acid scaffold from the *E. coli* EC (PDB ID 6ALH) transferred onto the RNAP-δ-HelD complex (HelD omitted) by superpositioning of the β subunits, showing competition of δ$^{CTR}$ with the downstream DNA duplex in the main channel. **d** Comparison to an *E. coli* σ$^{34}$ holoenzyme structure (PDB ID 6P1K), showing analogous positioning of δ$^{CTR}$ and the σ$^{1.1}$ globular domain in the main channel and the reduced channel width in the σ$^{34}$ holoenzyme. σ$^{34}$, sand-colored; σ$^{1.1}$, orange. **e** Comparison to an *M. smegmatis* σ$^{A}$ holoenzyme structure (PDB ID 6EYD), showing analogous positioning of δ$^{CTR}$ and a σ$^{1.1–1.2}$ linker helix in the main channel and the reduced channel width in the σ$^{A}$ holoenzyme. σ$^{A}$, sand-colored; σ$^{1.1–1.2}$ linker helix, orange. **f** Close-up view of RNAP active site region in RNAP-δ-HelD, with a nucleic acid scaffold from the *E. coli* EC (PDB ID 6ALH) transferred onto the RNAP-δ-HelD complex by superpositioning of the β subunits, illustrating direct competition of the HelD$^{NTR}$ coiled-coil tip with RNA (white arrow). **g** EMSA monitoring displacement of DNA (lanes 1–14) or DNA/RNA (lanes 15–28) from RNAP by HelD, δ or combinations. Top scheme, samples analyzed; gray boxes, a respective component added (proteins in equimolar amounts to RNAP$^{ΔδΔHelD}$). Numbers, molar ratios of δ or δ$^{NTD}$ relative to RNAP$^{ΔδΔHelD}$ added. Panels labeled "DNA" or "DNA/RNA", native PAGE analyses. nts, molecular weight marker (number of nucleotides). All lanes are from the same gel, some lanes for the DNA-only gel were removed for display purposes (dashed line). Bar graphs, quantification of the data shown in the middle panels. Values represent means of DNA or DNA/RNA bound relative to RNAP$^{ΔδΔHelD}$ alone ± SD (Excel, Microsoft Office Professional Plus 2016) for $n = 3$ independent experiments, using the same biochemical samples (data points indicated).

displacement emerged; however, due to the RNA-mediated stabilization of DNA on RNAP, HelD and δ individually or HelD/δ$^{NTD}$ liberated less RNAP, and higher concentrations of δ in the presence of HelD were required to achieve full nucleic acid displacement (Fig. 3g, lanes 15–28). Notably, δ/HelD-mediated DNA or DNA/RNA displacement did not require the addition of ATP. Together, these results explain why a nucleic acid scaffold failed to associate with the RNAP-δ-HelD complex during preparation for cryoEM; they underscore the importance of δ in nucleic acid displacement, show that HelD is required to achieve complete nucleic acid release and support the cooperation of δ$^{CTR}$ and HelD inferred from our structure and CLMS.

**ATP-dependent HelD release.** As HelD completely incapacitates RNAP (Fig. 5a, b), it has to be released to allow transcription to resume. σ$^{A}$ did not displace HelD in SEC (Supplementary Fig. 7a). Comparison of UvrD bound to DNA and ADP-Mg$_2$F$_3$[36] showed that the D1/D2 conformation of RNAP-bound HelD is incompatible with ATP binding (Fig. 6a). Since ATP binding to HelD induces conformational changes, as revealed by SAXS[27], we surmised that ATP-bound HelD may have a lower affinity for RNAP than the apo factor. Consistent with this notion, ATPγS, AMPPNP, and, to a somewhat lesser extent, ATP led to the release of HelD from RNAP-δ-HelD during SEC, while ADP or AMP had minor effects (Fig. 6b and Supplementary Fig. 7b). HelD exhibits intrinsic ATPase activity that is unaltered in the presence of RNAP[29]. Thus, AMPPNP and ATPγS mimic conditions of constantly high ATP supply, whereas ATP is likely hydrolyzed and separated from RNAP/HelD during SEC, reducing its effect. Unlike HelD, δ is not displaced from RNAP by the addition of ATP or analogs (Fig. 6b and Supplementary Fig. 7b).

**Dimeric (RNAP-δ-HelD)$_2$.** About two-thirds of our cryoEM particle images conformed to dimeric (RNAP-δ-HelD)$_2$ complexes (Fig. 6c), which were partially stable during SEC under conditions identical to cryoEM sample preparation (0.15% *n*-octylglucoside; Supplementary Fig. 1d). We also conducted negative stain EM analyses with RNAP-δ-HelD in the presence or absence of 0.15% *n*-octylglucoside and detected dimers under both conditions (Supplementary Fig. 1e; a quantitative analysis of the monomer/dimer distribution was precluded by preferred particle orientations on the carbon films). The protomers of the dimeric assembly closely resemble the monomeric RNAP-δ-HelD complex (root-mean-square deviation of 1.2–1.3 Å for 23,360–23,971 pairs of Cα atoms), but elements of the RNAP active site are further remodeled in the dimer (Fig. 5a, b). The

HelD-repositioned clamp forms an essential contact region in the dimer, which also involves the initiation/elongation factor-binding β flap tip (FT; Fig. 6c). The dimeric (RNAP-δ-HelD)$_2$ complex shows a striking resemblance to the hibernating dimeric eukaryotic RNAP I[41–43], with analogous regions contributing to the dimer interfaces (Fig. 6d). As in (RNAP-δ-HelD)$_2$, each protomer of the hibernating RNAP I dimer exhibits a wide-open DNA-binding cleft, partially unfolded bridge helix, and a DNA-mimicking loop stably bound inside the cleft[41–43], similar to δ$^{CTR}$. Furthermore, the A12.2 C-terminal domain of RNAP I is located inside the secondary channel[42]. These observations suggest that, like the RNAP I dimer, dimeric RNAP-δ-HelD may represent a dormant state.

## Discussion

Results of this and the accompanying reports[44,45] show that HelD mounts a two-pronged attack at the RNAP main and secondary channels. Both *B. subtilis* and the distantly related *M. smegmatis* HelD pinch RNAP around the BH, widen the main and RNA exit channels to provide escape routes for DNA and RNA, and displace the bound nucleic acids. However, the exact implementations of this conserved mechanism are distinct. *B. subtilis* HelD uses similarly sized arms to penetrate deeply into the channels, with δ playing a supporting role. δ$^{NTD}$ aids the main channel opening, whereas δ$^{CTR}$ may support HelD recruitment and guide HelD$^{Bumper}$ into the main channel to avoid topological trapping of DNA. In contrast, *M. smegmatis* HelD has evolved a branched main channel arm that functionally compensates for the absence of δ and for a rudimentary secondary channel arm, which merely helps HelD anchoring on RNAP. As HelD and δ did not require ATP addition to displace nucleic acids from RNAP, we presume that the large surface area buried upon RNAP-δ-HelD complex formation, rather than HelD ATPase, provides the driving force for the marked RNAP opening.

To engage RNAP, HelD reaches around the β2 lobe, a mode of attack that is not possible with RNAPs containing a β′ lineage-specific insertion, SI3, stacked onto the β2 lobe, such as *E. coli* (Supplementary Fig. 8a). Consistently, *E. coli* does not encode HelD, and a distantly related ATPase, RapA, has been proposed to aid RNAP recycling[15]. Unlike HelD, RapA binds near the RNA exit tunnel and does not induce major conformational changes in the EC (Supplementary Fig. 8b). Instead, RapA is thought to rescue ECs by promoting backtracking[46]. Alternative recycling mechanisms likely exist in SI3-containing species. Indeed, *E. coli* DksA has recently been proposed to remove RNAP from nucleic acids[47]. DksA binds in the secondary channel using a Gre-like coiled-coil[48], induces conformational changes in RNAP[49], albeit

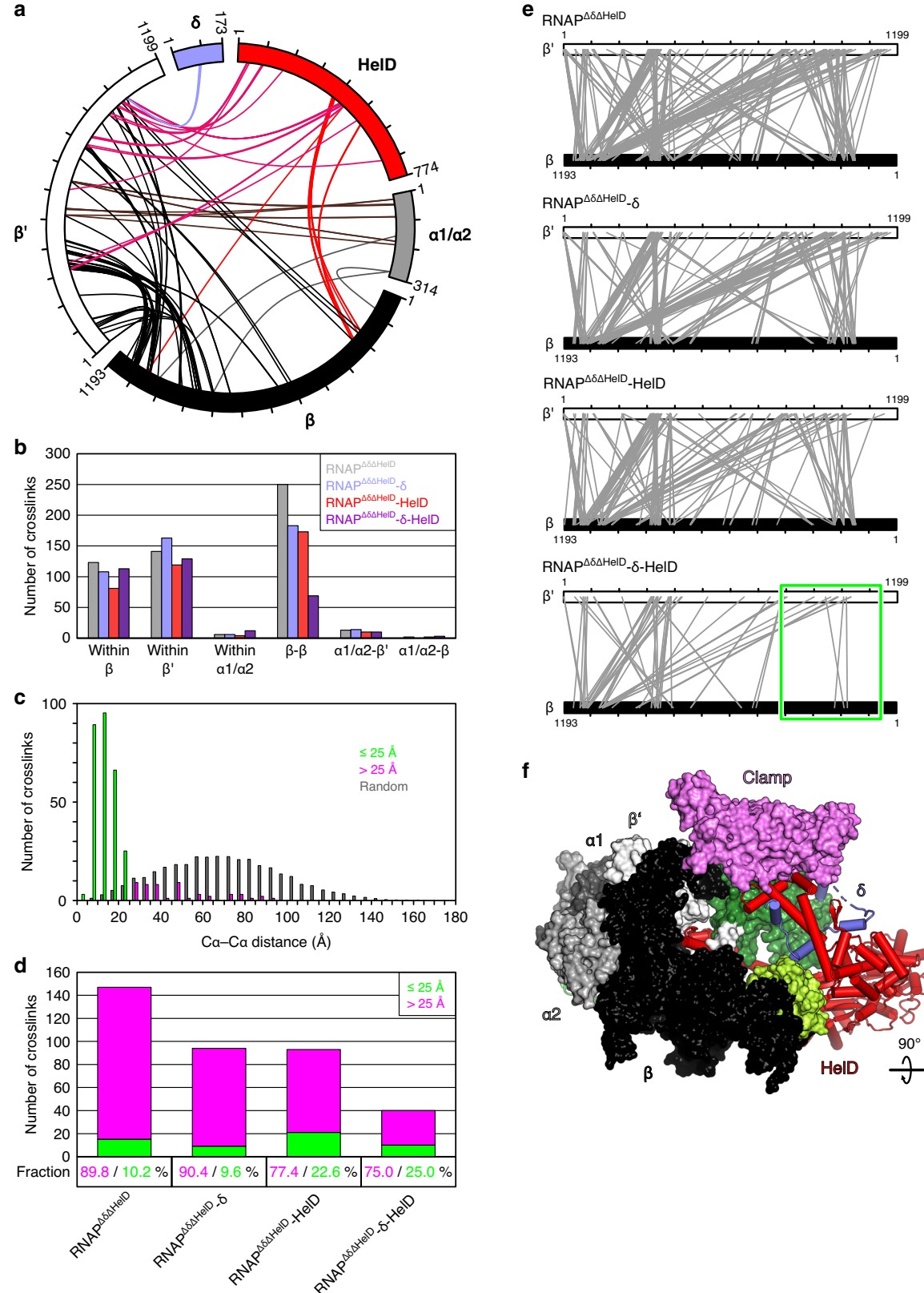

less marked than HelD, and is present only in bacteria that have SI3[50].

Our cryoEM structures also inform about likely mechanisms of action of the δ subunit during initiation and elongation. Previously, δ alone had been shown to displace nucleic acids from RNAP[7], a result we recapitulate here (Fig. 3g). As δ^CTR peptides

showed similar activity when added in excess and as δ^NTD was found to bind RNAP, δ-mediated nucleic acid displacement was suggested to involve δ^NTD-dependent tethering of the polyanionic δ^CTR to core RNAP[7]. Our cryoEM structures confirm and further refine this hypothesis. δ^NTD anchors δ^CTR at the rim of the main channel; due to its length and intrinsic disorder, δ^CTR can reach

**Fig. 4 Structure probing by CLMS. a** Map of hetero-protein crosslinks observed in RNAP$^{\Delta\delta\Delta HelD}$-δ-HelD complex. **b** Crosslinks identified in RNAP$^{\Delta\delta\Delta HelD}$, RNAP$^{\Delta\delta\Delta HelD}$-δ, RNAP$^{\Delta\delta\Delta HelD}$-HelD, and RNAP$^{\Delta\delta\Delta HelD}$-δ-HelD. Binding of both δ and HelD leads to strongly reduced crosslinking between β and β'.
**c** Distribution of Cα-Cα distances between crosslinked residue pairs in reference to the RNAP-δ-HelD structure. Crosslinks with Cα-Cα distances within 25 Å, the theoretical crosslinking limit of sulfo-SDA, green; crosslinks with Cα-Cα distances >25 Å, magenta; distance distribution of random residue pairs in the RNAP-δ-HelD structure, gray. **d** Numbers of crosslinks (bars) between β and β' identified from the four cross-linked complexes, and fractions of over-length crosslinks (percentages at the bottom). Crosslinks are color-coded as in **b**. In the RNAP$^{\Delta\delta\Delta HelD}$-δ-HelD complex, a significantly reduced number of β-β' over-length crosslinks (in reference to the RNAP-δ-HelD structure) compared to the RNAP$^{\Delta\delta\Delta HelD}$, RNAP$^{\Delta\delta\Delta HelD}$-δ, and RNAP$^{\Delta\delta\Delta HelD}$-HelD complexes suggests that δ and HelD cooperate to stabilize an open conformation of RNAP. **e** Comparison of β-β' crosslinks observed with RNAP$^{\Delta\delta\Delta HelD}$, RNAP$^{\Delta\delta\Delta HelD}$-δ, RNAP$^{\Delta\delta\Delta HelD}$-HelD, and RNAP$^{\Delta\delta\Delta HelD}$-δ-HelD. The green boxed region, crosslinks between the β1/2 lobes (residues 146–248) and the β' shelf and jaw (residues 794–1141) observed in the first three complexes but almost absent in RNAP$^{\Delta\delta\Delta HelD}$-δ-HelD. **f** Structure of the RNAP-δ-HelD complex highlighting the β1/2 lobes (lemon green) and β' shelf and jaw (forest green), which largely lack crosslinks in the presence of δ and HelD (green box in **e**).

into and explore most of the volume of the main channel, spatially and electrostatically competing with bound nucleic acids. This mode of action would explain how δ enhances core RNAP recycling in multi-round assays[7,22], and it may constitute the main recycling mechanism in bacteria that contain δ but lack HelD. It also provides an explanation for the finding that in vitro, RNAP-δ in the presence of σ factors still binds promoters and forms closed complexes but fails to establish contacts with the downstream DNA[22,51,52], which are required for the transition to an open complex. Finally, the model suggests that abolishing a positively charged region at the δ$^{CTR}$ N-terminus, to promote more extended conformations of the CTR[23], effectively abrogates a restraint on δ$^{CTR}$'s ability to invade the main channel, thus reconciling increased effects of such CTR variants at promoters that form unstable complexes[23].

δ exhibits negative cooperativity with σ$^A$ and favors its exchange for alternative σ factors that lack σ$^{1.1}$ [17,53]. In the E. coli σ$^{34}$ holoenzyme, σ$^{1.1}$ can reside in the main channel, preventing access of either double- or single-stranded DNA to the RNAP active site[34] (Fig. 3d). To allow for DNA loading, the clamp has to open further[54] or σ$^{1.1}$ has to move[55]. These observations suggest binding competition between δ and σ$^{1.1}$, fully in line with our structures (Fig. 3c–e). However, while δ$^{NTD}$ resembles the globular domain of σ$^{1.1}$ [32], our results indicate that the structurally unrelated CTR (together with HelD$^{Bumper}$, if present) constitutes the σ$^{1.1}$-competitive element that can occupy equivalent regions in the main channel (Fig. 3c).

The HelD/δ-dependent recycling mechanism uncovered here represents a marvelously simple, direct, and effective way of recovering RNAP from virtually any state trapped post-termination. However, RNAP is truly recycled only when HelD also detaches. We show that HelD is released by ATP (Fig. 6b and Supplementary Fig. 7b), suggesting that high levels of ATP could help prevent HelD from trapping RNAP in an inactivated complex during exponential growth. Noteworthy, both B. subtilis and M. smegmatis HelDs cannot bind ATP when fully engaged with RNAP, suggesting that intrinsically timed isomerization into a less engaged conformation must precede ATP binding and release from RNAP. Irrespectively, we suggest that ATP-mediated HelD release underlies the ATP-dependent stimulatory effect of HelD on transcription[29]. In contrast, ATP does not induce the concomitant release of δ (Fig. 6b and Supplementary Fig. 7b), confirming that δ has an intrinsically high affinity for RNAP and does not require HelD to remain stably associated. As an association of alternative σ factors (relative to σ$^A$) is favored in RNAP-δ compared to RNAP lacking δ[17,53], additional mechanisms may be at play to remove δ (or expunge δ$^{CTR}$ from the main channel) in situations where efficient rebinding of σ$^A$ is specifically required.

When cells sporulate during the stationary phase, conversely, the levels of ATP are low[56], transcription is limited, HelD levels

match those of RNAP[30], and HelD is thus expected to remain bound to RNAP. Given that HelD locks RNAP in an inactive state, could it be used to store RNAP until the conditions improve? Intriguingly, we observed (RNAP-δ-HelD)$_2$ dimers resembling hibernating eukaryotic RNAP I (Fig. 6c, d), which were partially stable in SEC at initial RNAP concentrations about 10-fold lower compared to their nominal cellular concentrations in the log phase, estimated from transcript levels and ribosome profiling[30,57]. Dimerization of RNAP has also been reported in bacteria that lack HelD, including E. coli[9]. While dimerization may be an inherent property of RNAPs, our results clearly show that HelD, while not directly involved in forming the dimer interface, facilitates the observed mode of dimerization by pushing the β' clamp outwards to enable homologous contacts between the β' clamps, the C-terminal β clamp, and regions of the β flap (Fig. 6c). Notably, a comparison of our dimeric structure to an M. smegmatis RNAP-σ$^A$ holoenzyme structure[38] shows that all binding sites for σ, except for σ$^{1.1}$ in the main channel, would be accessible in the RNAP-δ-HelD dimer. Thus, rebinding of σ could contribute to the efficient recovery of RNAP from the dimeric state. Taken together, HelD/δ could in principle promote RNAP hibernation that may be essential for fast RNAP recovery, in line with observations that overexpression of HelD enhances sporulation[58] and deletions of HelD, δ or both prolong the lag phase[29]. Further tests of this idea are required and could involve in vivo CLMS at different growth phases and during sporulation, in WT compared to ΔhelD or overexpressing cells, or in vivo super-resolution imaging with fluorescence labeling of HelD or RNAP.

This and the accompanying studies present a hitherto unrecognized transcription recycling system that safeguards genome integrity and may contribute to persistence during periods of dormancy. In our model (Fig. 7), parts of which require further validation, reservoirs of active RNAP are controlled by HelD, which may rescue trapped RNAP during fast growth, promote RNAP hibernation during slow growth, and enable efficient RNAP recovery upon shift to a nutrient-rich environment. We note that although most laboratory experiments are carried out with rapidly growing bacteria for convenience, dormant states are prevalent in natural environments and pose grave health risks. For example, B. anthracis spores are the infectious particles for anthrax, whereas slow-growing Pseudomonas aeruginosa biofilms and M. tuberculosis are resistant to cidal antibiotics. Unraveling the regulation of dormancy is thus critical for the understanding of bacterial physiology and identifying new strategies for the eradication of multidrug-resistant pathogens.

## Methods
**Plasmids, DNAs, and RNAs**. A DNA fragment encoding B. subtilis HelD was PCR-amplified from strain MH5636 (Supplementary Table 1). The PCR product

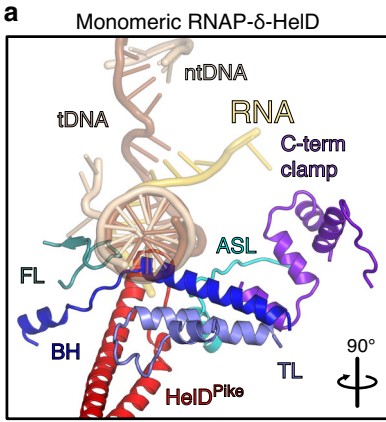

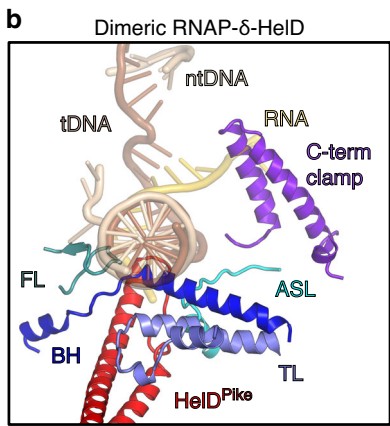

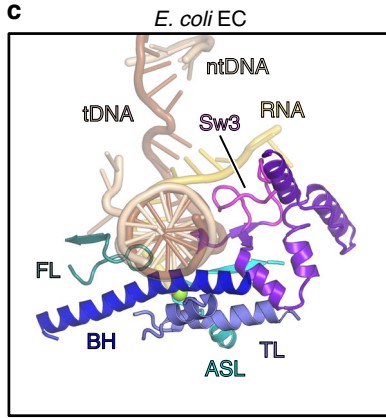

**Fig. 5 Active site dismantling. a**, **b** RNAP active site environments in monomeric RNAP-δ-HelD (**a**) and dimeric (**b**) RNAP-δ-HelD, showing HelD-mediated active site dismantling. Comparison to Newing et al.[44] suggests that the presence of δ promotes more severe active site penetration by HelD[Pike]. Nucleic acids of an *E. coli* EC (PDB ID 6ALH) were transferred to RNAP-δ-HelD by superposition of the β subunits. β elements: FL (fork loop), teal; C-term clamp (C-terminal clamp), purple; Sw3, magenta. β′ elements: ASL, cyan; BH (bridge helix), blue; TL (trigger loop), slate blue. The catalytic Mg$^{2+}$ ion (green sphere) is lost from RNAP-δ-HelD upon HelD[Pike] invasion. **c** Comparison to the RNAP active site environment in an *E. coli* EC (PDB ID 6ALH).

was inserted into expression vector pGEX-6p-1 via *BamH*I and *Xho*I restriction sites, in frame with a region encoding an N-terminal GST-tag. DNA fragments encoding *B. subtilis* σ$^A$, δ or δ$^{NTD}$ were PCR-amplified from strain MH5636 and inserted into a pETM-11 vector (EMBL, Heidelberg) via *Nco*I/*Hind*III or *Nco*I/*Xho*I restriction sites, respectively, in frame with a region encoding an N-terminal His$_6$-tag. DNA and RNA oligomers used for the assembly of transcription complexes were purchased from Eurofins and IBA Lifesciences, respectively.

**Protein production and purification.** *B. subtilis* strains MH5636, LK782 (Δ*helD*) or LK1032 (Δ*helD*Δ*rpoE*; Supplementary Table 1) were used to produce stationary phase RNAP, RNAP$^{ΔHelD}$ or RNAP$^{ΔΔHelD}$, respectively. In these strains, the chromosomally-encoded β′ subunit carries a C-terminal His$_{10}$-tag. Strains were grown in TB medium at 37 °C to an OD$_{600}$ of 1.0 and were then shifted to 18 °C and grown to an OD$_{600}$ of about 11. All purification steps were performed at 4 °C. Cells were harvested by centrifugation, resuspended in buffer A (50 mM Na$_2$HPO$_4$, 300 mM NaCl, 3 mM 2-mercaptoethanol, 5% [v/v] glycerol, pH 7.9), and lysed by sonication. The lysate was cleared by centrifugation. RNAP variants were captured on Ni$^{2+}$-NTA affinity resin (Macherey-Nagel), washed with buffer A supplemented with 25 mM imidazole, and eluted with buffer A supplemented with 250 mM imidazole. The eluate was dialyzed overnight against 50 mM Na$_2$HPO$_4$, 300 mM NaCl, 3 mM DTT, 5% [v/v] glycerol, pH 7.9, loaded on a 5 ml HiTrap Heparin HP column (GE Healthcare), washed with buffer B (50 mM TRIS-HCl, 100 mM NaCl, 3 mM DTT, 0.1 mM EDTA, 5% [v/v] glycerol, pH 7.9) and eluted with a linear gradient to buffer B with 700 mM NaCl. Fractions containing RNAPs were pooled and further purified by SEC on a HiLoad Superdex 200 Increase 16/600 column (GE Healthcare) in 20 mM TRIS-HCl, 150 mM NaCl, 0.5 mM DTT, 5% (v/v) glycerol, pH 8.0. The final samples were concentrated to approximately 16 mg/ml. RNAP produced from strain MH5636 was directly used for EM sample preparation. Other RNAP preparations were aliquoted, flash-frozen in liquid N$_2$, and stored at −80 °C.

Recombinant *B. subtilis* GST-HelD was produced in *E. coli* Rosetta(DE3) cells, His$_6$-δ, His$_6$-δ$^{NTD}$, and His$_6$-σ$^A$ were produced in *E. coli* BL21(DE3)-RIL cells. Cells were grown in auto-inducing media[59] at 37 °C to an OD$_{600}$ of 1.0 and further incubated at 20 °C overnight. All purification steps were performed at 4 °C. GST-HelD cells were harvested by centrifugation, resuspended in buffer C (50 mM TRIS-HCl, 500 mM NaCl, 1 mM 2-mercaptoethanol, 10% [v/v] glycerol, pH 7.9), and lysed by sonication. The lysate was cleared by centrifugation, GST-HelD was captured on glutathione resin (Macherey-Nagel), washed with buffer C, and eluted with 50 mM TRIS-HCl, 300 mM NaCl, 1 mM DTT, 10% (v/v) glycerol, 20 mM reduced glutathione, pH 7.9. Eluted fractions were dialyzed against buffer D (20 mM TRIS-HCl, 200 mM NaCl, 1 mM DTT, 5% [v/v] glycerol, pH 7.9) in the presence of GST-tagged PreScission protease. HelD was separated from uncleaved protein, GST, and GST-PreScission by a second passage through glutathione resin. The flowthrough was further purified by SEC on a HiLoad Superdex 200 Increase 16/600 column equilibrated in buffer D. Fractions containing HelD were concentrated to approximately 15 mg/ml, aliquoted, flash-frozen in liquid N$_2$, and stored at −80 °C.

His$_6$-δ or His$_6$-δ$^{NTD}$ cells were harvested by centrifugation, resuspended in 50 mM TRIS-HCl, 500 mM NaCl, 0.5 mM 2-mercaptoethanol 5% [v/v] glycerol, pH 6.0, and lysed by sonication. The lysate was cleared by centrifugation, His$_6$-δ/His$_6$-δ$^{NTD}$ was captured on Ni$^{2+}$-NTA resin, washed with 50 mM TRIS-HCl, 300 mM NaCl, 0.5 mM 2-mercaptoethanol, 10 mM imidazole, 5% (v/v) glycerol, pH 6.0, and eluted with 20 mM TRIS-HCl, 150 mM NaCl, 0.5 mM 2-mercaptoethanol, 400 mM imidazole, 5% (v/v) glycerol, pH 6.0. For the assembly of complexes for cryoEM analysis, eluted His$_6$-δ was supplemented with His-tagged TEV protease (1:40 [w/w]), dialyzed against buffer E (20 mM TRIS-HCl, 150 mM NaCl, 1 mM DTT, 5% (v/v) glycerol, pH 6.0) overnight and passed through fresh Ni$^{2+}$-NTA resin to remove the uncleaved His$_6$-δ, the cleaved His$_6$-tag, and His-tagged TEV protease. Proteins were further purified by SEC on a Superdex 75 Increase 10/300 column (GE Healthcare) in buffer E. Fractions containing His$_6$-δ, δ or His$_6$-δ$^{NTD}$ were concentrated to ~4 mg/ml (His$_6$-δ, His$_6$-δ$^{NTD}$) and 23 mg/ml (δ), aliquoted, flash-frozen in liquid N$_2$ and stored at −80 °C.

σ$^A$ cells were harvested by centrifugation, resuspended in buffer F (20 mM TRIS-HCl, 500 mM NaCl, 1 mM 2-mercaptoethanol, 5% [v/v] glycerol, pH 7.5) supplemented with 20 mM imidazole, and lysed by sonication. The lysate was cleared by centrifugation, His$_6$-σ$^A$ was captured on Ni$^{2+}$-NTA resin, washed with buffer F supplemented with 50 mM imidazole, and eluted with buffer F supplemented with 400 mM imidazole. Eluted His$_6$-σ$^A$ was supplemented with His-tagged TEV protease (1:40 [w/w]), dialyzed against buffer F supplemented with 1 mM EDTA overnight, and passed through fresh Ni$^{2+}$-NTA resin to remove uncleaved His$_6$-σ$^A$, cleaved His$_6$-tag and His-tagged TEV protease. The target protein was further purified by SEC on a Superdex 75 Increase 16/600 column (GE Healthcare) in 25 mM TRIS-HCl, 300 mM NaCl, 0.1 mM DTT, 5% (v/v) glycerol, pH 7.5. Fractions containing σ$^A$ were concentrated to approximately 39 mg/ml, aliquoted, flash-frozen in liquid N$_2$, and stored at −80 °C.

**Crosslinking/mass spectrometry.** Sulfo-SDA predominantly establishes lysine-X crosslinks through a primary amine-reactive moiety on one side and a UV-activatable moiety on the other (theoretical crosslinking limit 25 Å). Sulfo-SDA was prepared at 3 mg/ml in 20 mM HEPES-NaOH, 5 mM Mg(OAc)$_2$, 300 mM NaCl, 5 mM DTT, 5% (v/v) glycerol, pH 8.0 immediately prior to addition of RNAP$^{Δδ-ΔHelD}$, RNAP$^{ΔδΔHelD}$-δ, RNAP$^{ΔδΔHelD}$-HelD, or RNAP$^{ΔδΔHelD}$-δ-HelD (protein: sulfo-SDA 1:3 [w/w]). Samples were incubated on ice for two hours and then irradiated in a thin film using 365 nm UV (UVP CL-1000 UV Crosslinker, UVP Inc.) for 20 min on ice (5 cm distance from UV-A lamp). The cross-linked samples were separated by 4–12% BIS-TRIS NuPAGE, gel bands corresponding to cross-linked monomeric complexes were excised and digested in-gel[60]. The resulting peptides were desalted using C18 StageTips[61].

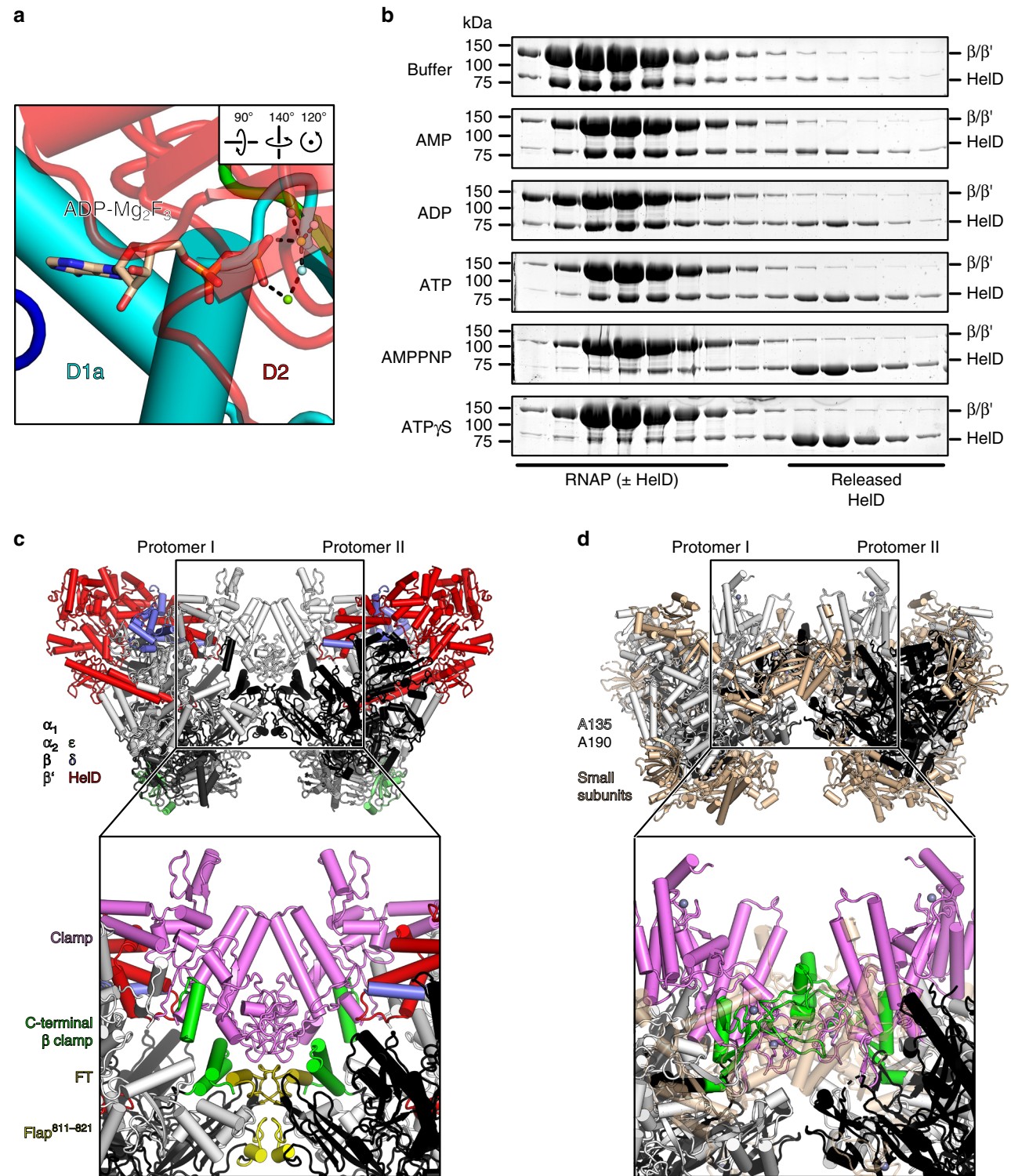

**Fig. 6 HelD release and RNAP-HelD complex dimerization. a** Close-up view of the ATP-binding site of HelD, with ADP-Mg$_2$F$_3$ from a UvrD complex (PDB ID 2IS6) transferred by superpositioning of the UvrD NTPase domains on HelD, illustrating clashes with the nucleotide. ADP-Mg$_2$F$_3$ shown as sticks and colored by atom type; carbon, beige; nitrogen, blue; oxygen, red; phosphorus, orange; magnesium ions, green; fluoride ions, light blue. **b** SDS-PAGE analysis of SEC runs after treating RNAP-δ-HelD with buffer or the nucleotides indicated on the left. Experiments were repeated independently at least twice with similar results. For full gels, see Supplementary Fig. 7b. **c** Structure of dimeric RNAP-δ-HelD. Inset, close-up view on the dimer interface. The two protomers interact via the elements highlighted in colors; β' clamp, violet; C-terminal β clamp, green; β flap tip (FT), olive; residue 811–821 of the β flap (Flap[811–821]), yellow. **d** Structure of a hibernating RNAP I dimer (PDB ID 4C2M). A135 subunit, black; A190 subunit, white; small subunits, beige. Inset, close-up view on the dimer interface. A190 clamp, violet; C-terminal A135 clamp, green.

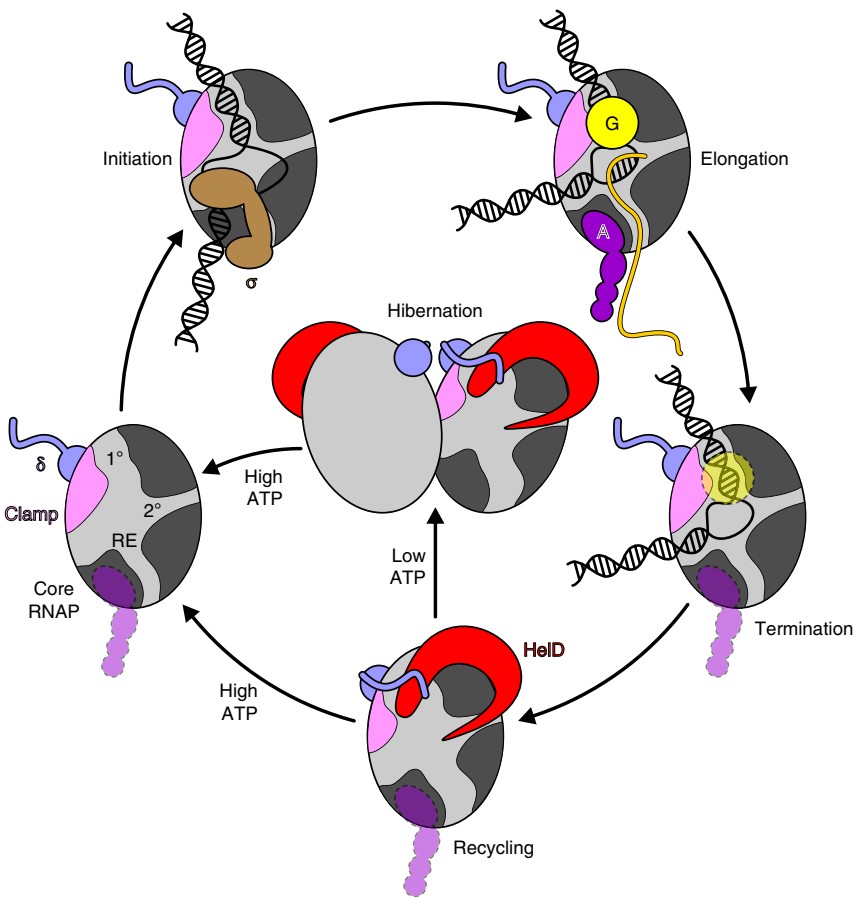

**Fig. 7 Model for HelD/δ-mediated RNAP recycling and putative hibernation.** 1°/2°, main/secondary channels; RE, RNA exit tunnel; A/G, general elongation factors NusA/NusG. NusG binds across the active center cleft, while NusA binds to the β FT. Semi-transparent icons with dashed lines indicate that the respective factor may be released at the respective step. If the factors remain after termination, NusG will likely be displaced by HelD-induced main channel opening, while the NusA binding site is sequestered in dimeric (RNAP-δ-HelD)₂. Hibernation by RNAP-δ-HelD dimerization and ATP-mediated recovery from the dormant state represent tentative aspects of the model.

10% of each sample were analyzed by LC-MS/MS without fractionation, the remaining 90% were fractionated using SEC on a Superdex Peptide 3.2/300 column (GE Healthcare) in 30% (v/v) acetonitrile, 0.1% (v/v) trifluoroacetic acid at a flow rate of 10 μl/min to enrich for crosslinked peptides[62]. The first six peptide-containing fractions (50 μl each) were collected, the solvent was removed using a vacuum concentrator and the fractions were analyzed by LC-MS/MS on an Orbitrap Fusion Lumos Tribrid mass spectrometer (ThermoFisher Scientific), connected to an Ultimate 3000 RSLCnano system (Dionex, ThermoFisher Scientific).

The non-fractionated samples were injected onto a 50 cm EASY-Spray C18 LC column (ThermoFisher Scientific) operated at 50 °C. Peptides were separated using a linear gradient going from 2% mobile phase B (80% [v/v] acetonitrile, 0.1% [v/v] formic acid) to 40% mobile phase B in mobile phase A (0.1% [v/v] formic acid) at a flow rate of 0.3 μl/min over 110 min, followed by a linear increase from 40 to 95% mobile phase B in 11 min. Eluted peptides were ionized using an EASY-Spray source (ThermoFisher Scientific) and MS data were acquired in the data-dependent mode with the top-speed option. For each 3-s acquisition cycle, the full scan mass spectrum was recorded in the Orbitrap with a resolution of 120,000. The ions with a charge state from 3+ to 7+ were isolated and fragmented using higher-energy collisional dissociation (HCD) with 30% collision energy. The fragmentation spectra were then recorded in the Orbitrap with a resolution of 50,000. Dynamic exclusion was enabled with a single repeat count and 60 s exclusion duration.

SEC fractions were analyzed using an identical LC-MS/MS setup. Peptides were separated by applying a gradient ranging from 2 to 45% mobile phase B (optimized for each fraction) over 90 min, followed by ramping up mobile phase B to 55 and 95% within 2.5 min each. For each three-second data-dependent MS acquisition cycle, the full scan mass spectrum was recorded in the Orbitrap with a resolution of 120,000. The ions with a charge state from 3+ to 7+ were isolated and fragmented using HCD. For each isolated precursor, one of three collision energy settings (26%, 28%, or 30%) was selected for fragmentation using a data-dependent decision tree based on the *m/z* and charge of the precursor. The fragmentation spectra were recorded in the Orbitrap with a resolution of 50,000. Dynamic exclusion was enabled with a single repeat count and 60 s exclusion duration.

LC-MS/MS data generated from the four complexes were processed separately. MS2 peak lists were generated from the raw MS data files using the MSConvert module in ProteoWizard (version 3.0.11729). The default parameters were applied, except that Top MS/MS Peaks per 100 Da was set to 20 and the denoising function was enabled. Precursor and fragment *m/z* values were recalibrated. Identification of cross-linked peptides was carried out using xiSEARCH software (https://www.rappsilberlab.org/software/xisearch; version 1.7.4)[63]. For RNAP^ΔδΔHelD, peak lists were searched against the sequence and the reversed sequence of RNAP subunits (α, β, β', and ε) and two co-purified proteins, σ^A and σ^B. For RNAP^ΔδΔHelD-δ, RNAP^ΔδΔHelD-HelD, and RNAP^ΔδΔHelD-δ-HelD samples, protein sequences of δ, HelD or both were included in the database. The following parameters were applied for the search: MS accuracy = 4 ppm; MS2 accuracy = 8 ppm; enzyme = trypsin (with full tryptic specificity); allowed number of missed cleavages = 2; missing monoisotopic peak = 2; crosslinker = sulfo-SDA (the reaction specificity for sulfo-SDA was assumed to be for lysine, serine, threonine, tyrosine, and protein N-termini on the NHS ester end, and any amino acid residue for the diazirine end); fixed modifications = carbamidomethylation on cysteine; variable modifications = oxidation on methionine and sulfo-SDA loop link. Identified crosslinked peptide candidates were filtered using xiFDR[64]. A false discovery rate of 5% on the residue-pair level was applied with the "boost between" option selected. Crosslinked residue pairs identified from the four complexes are summarized in Supplementary Table 4 and Supplementary Data 1.

**CryoEM sample preparation, data collection, and processing**. Equimolar amounts of tDNA, ntDNA, and RNA were mixed in buffer G (20 mM TRIS-HOAc, 5 mM Mg[OAc]₂, 100 mM KOAc, 2 mM DTT, 5% [v/v] glycerol, pH 8.0) and annealed by heating to 95 °C for 5 min and subsequent cooling to 25 °C at 1 °C/min. The annealed scaffold was incubated with *B. subtilis* RNAP in a 1.3:1 molar ratio in buffer H (20 mM TRIS-HOAc, 5 mM Mg[OAc]₂, 300 mM KOAc, 2 mM DTT, 5% [v/v] glycerol, pH 8.0) for 10 min on ice, then for 10 min at 32 °C. Equimolar amounts (to RNAP) of δ and HelD were added stepwise, followed by incubation for 10 min at 32 °C after each addition. The mixture was subjected to

SEC on a Superdex 200 Increase 3.2/300 column (GE Healthcare) in buffer H. Fractions containing RNAP, δ, and HelD were pooled and concentrated to approximately 5 mg/ml.

Immediately before preparation of the grids, the sample was supplemented with 0.15% (w/v) n-octylglucoside (critical micellar concentration 0.6% [w/v]). 3.8 μl of the final mixture were spotted on plasma-treated Quantifoil R1/2 holey carbon grids at 10 °C/100% humidity and plunged into liquid ethane using an FEI Vitrobot Mark IV. Image acquisition was conducted on an FEI Titan Krios G3i (300 kV) transmission electron microscope (TEM) with a Falcon 3EC camera at a nominal magnification of 92,000x in counting mode using EPU software (ThermoFisher Scientific) with a calibrated pixel size of 0.832 Å. A total electron dose of 40 $e^-/Å^2$ was accumulated over an exposure time of 36 s. Movie alignment was done with MotionCor2[65] using 5 × 5 patches followed by ctf estimation with Gctf[66].

All following image analysis steps were done with cryoSPARC[67]. Class averages of manually selected particles were used to generate an initial template for reference-based particle picking from 9127 micrographs. Particle images were extracted with a box size of 440 and binned to 110 for initial analysis. Ab initio reconstruction using a small subset of particles was conducted to generate an initial 3D reference for 3D heterogeneous refinement. The dataset was iteratively classified into two well-resolved populations representing monomeric and dimeric RNAP-δ-HelD. Selected particles were re-extracted with a box of 220 and again classified in 3D to further clean the dataset. Finally, selected particle images were re-extracted with a box of 280 (1.3 Å/px) and subjected to local refinement using a generously enlarged soft-mask for monomeric or dimeric RNAP-δ-HelD. Local refinement of the dimer particles using the monomeric mask was conducted as a control to trace differences of RNAP-δ-HelD in the authentic monomer and dimer structures. After per-particle CTF correction, non-uniform refinement was applied to generate the final reconstructions.

**Negative stain EM analysis**. RNAP-δ-HelD complex was prepared as for cryoEM analysis, diluted to 25 μg/ml in buffer H and supplemented with 0.15% n-octylglucoside or buffer immediately before grid preparation. 4 μl of the samples were added to glow-discharged Formvar/carbon grids (S162, Plano GmbH), left to settle for 40 s and manually blotted with Whatman paper No. 1, followed by addition of 4 μl of 1% (w/v) uranyl acetate staining solution. After 40 s incubation, the grids were manually blotted and dried at ambient temperature overnight. Samples were imaged on an FEI Talos L120C TEM, operated at 120 kV, equipped with an FEI CETA 16 M CCD camera at a nominal magnification of 57,000x. The calibrated pixel size was 2.53 Å/px. Images were acquired manually in low dose mode using TEM Imaging & Analysis (TIA) software, supplied by the manufacturer, accumulating a total electron dose of 50 $e^-/Å^2$. Image analysis was done with cryoSPARC. After CTF estimation, manually selected particle images were used as a reference for template-based particle picking. Particle images were extracted with a box size of 160 px and resampled to 80 px. A mask of 220 Å diameter was applied during 2D classification.

**Model building and refinement**. The final cryoEM map for the dimeric RNAP-δ-HelD complex was used for initial model building. Coordinates of M. smegmatis RNAP α, β, and β′ subunits (PDB ID 5VI8)[68] were docked into the cryoEM map using Coot[69]. Modeling of δ was based on the NMR structure of B. subtilis δ (PDB ID 2M4K)[70]. Modeling of ε was supported by the structure of YkzG from Geobacillus stearothermophilus (PDB ID 4NJC)[26]. Model building of HelD was supported by the structure of UvrD helicase from E. coli (PDB ID 3LFU)[71] as well as the C-terminal domain of a putative DNA helicase from Lactobacillus plantarun (PDB ID 3DMN). The subunits were manually rebuilt into the cryoEM map. The model was completed and manually adjusted residue-by-residue, supported by real-space refinement in Coot. The manually built model was refined against the cryoEM map using the real-space refinement protocol in PHENIX[72]. Model building of the monomeric complex was done in the same way but starting with a model of half of the dimeric complex. The structures were evaluated with Molprobity[73]. Structure figures were prepared using PyMOL (Version 1.8 Schrödinger, LLC).

**Structure comparisons**. Structures were compared by global superposition of complex structures or by superposition of selected subunits in complexes using the "secondary structure matching" algorithm implemented in Coot or the "align" algorithm implemented in PyMOL.

**Size exclusion chromatography/multi-angle light scattering**. SEC/MALS analysis was performed on an HPLC system (Agilent) coupled to mini DAWN TREOS multi-angle light scattering and RefractoMax 520 refractive index detectors (Wyatt Technology). RNAP-δ-HelD complex was assembled as for cryoEM. 60 μl of the sample at 1–1.3 mg/ml were chromatographed on a Superose 6 Increase 10/300 column (GE Healthcare) in buffer H or buffer H plus 0.15% (w/v) n-octylglucoside, supplemented with 0.02% (w/v) NaN₃, at 18 °C with a flow rate of 0.6 ml/min. Data were analyzed with the ASTRA 6.1 software (Wyatt Technology) using monomeric bovine serum albumin (Sigma-Aldrich) as a reference.

**Interaction assays**. HelD interactions with δ or $δ^{NTD}$ were analyzed by analytical SEC. 21 μM HelD and 42 μM δ or $δ^{NTD}$ were mixed in 20 mM HEPES-NaOH, 50 mM NaCl, 1 mM DTT, pH 7.5, and incubated for 10 min at room temperature. 50 μl of the samples were loaded on a Superdex 200 Increase PC 3.2 column (GE Healthcare) and chromatographed at 4 °C with a flow rate of 40 μl/min. Fractions were analyzed by 12.5% SDS-PAGE.

**Nucleic acid displacement assays**. Equimolar amounts of 5′-[$^{32}$P]-labeled ntDNA and unlabeled tDNA capable of forming an artificial bubble, or additionally an RNA 9-mer with complementarity to the tDNA in the bubble (Supplementary Table 1), were mixed in buffer G and annealed by heating to 95 °C for 5 min and subsequent cooling to 25 °C at 1 °C/min. The labeled DNA duplex or DNA/RNA scaffold and RNAP$^{ΔδΔHelD}$ (10 nM and 1 μM final concentrations, respectively) were incubated in buffer G for 10 min at 4 °C, followed by an additional 10 min incubation at 32 °C. Subsequently, (i) buffer, (ii) HelD (1 μM final concentration); (iii) δ (1 μM final concentration), (iv) combinations of HelD (1 μM final concentration) and δ (titrated final concentration; Fig. 3g) or (v) HelD and $δ^{NTD}$ (1 μM final concentration each) were added, and the samples were further incubated for 10 min at 32 °C. Samples were loaded on a 4% native PAGE gel and electrophoresed in 0.5X TBE buffer. Radiolabeled bands were visualized using a Storm phosphorimager and quantified using ImageQuant software (GE Healthcare).

**HelD release assays**. Equimolar amounts of HelD and stationary phase RNAP were mixed in buffer I (20 mM TRIS-HCl, 300 mM NaCl, 2 mM DTT, 5% (v/v) glycerol, pH 8.0), incubated for 10 min on ice and then for 10 min at 32 °C. The sample was chromatographed on a HiLoad Superdex 200 Increase 10/300 column (GE Healthcare) in buffer I. Fractions were analyzed by 12.5% SDS-PAGE, fractions containing RNAP-HelD complex were collected and concentrated to ~3 mg/ml (6.7 μM). 80 μl of this complex were mixed with buffer I, 5 mM Mg$^{2+}$-ATPγS/AMPPNP/ATP/ADP/AMP, 6.7 μM σ$^A$ or σ$^A$ plus Mg$^{2+}$-ATPγS in buffer I. 90 μl of the samples were loaded on a Superdex 200 Increase PC 3.2 column (GE Healthcare) and chromatographed at 4 °C with a flow rate of 40 μl/min. Fractions were analyzed by 12.5% SDS-PAGE.

## Data availability

CryoEM maps have been deposited in the Electron Microscopy Data Bank (https://www.ebi.ac.uk/pdbe/emdb/) under accession codes EMD-11104 (monomeric RNAP-δ-HelD) and EMD-11105 (dimeric RNAP-δ-HelD). Structure coordinates have been deposited in the RCSB Protein Data Bank (https://www.rcsb.org/) with accession codes 6ZCA[74] (monomeric RNAP-δ-HelD) and 6ZFB[75] (dimeric RNAP-δ-HelD). CLMS data have been deposited in jPOST (https://jpostdb.org/) with accession code JPST000858 (PXID PXD019437)[76]. Other data are available from the corresponding author upon reasonable request. Source data are provided with this paper.

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

## Acknowledgements

We are grateful to Dr. John D. Helmann for valuable comments. *B. subtilis* strains MH5636, LK782 (ΔhelD), and LK1032 (ΔhelDΔrpoE) were a kind gift from Libor Krásný, Czech Academy of Sciences. We thank Nicole Dimos, Freie Universität Berlin, for help in RNAP purification, Christoph Weise, Freie Universität Berlin, for MS-based finger-printing, and Boris Schade and Benedikt Kirmayer for assistance in cryoEM. We acknowledge access to electron microscopic equipment at the core facility BioSupraMol of Freie Universität Berlin, supported through grants from the Deutsche For-schungsgemeinschaft and the state of Berlin for large equipment according to Art. 91b GG (INST 335/588-1 FUGG, INST 335/589-1 FUGG, INST 335/590-1 FUGG). We are grateful for access to high-performance computing resources at the Zuse Institut Berlin. This work was supported by grants from the Deutsche Forschungsgemeinschaft (SFB 973/C08 and 433623608 to M.C.W.; GRK 2473 to M.C.W. and J.R.), the National Institutes of Health (GM067153 to I.A.), the Sigrid Jusélius Foundation (to G.A.B.), and by the Wellcome Trust through a Senior Research Fellowship to J.R. (103139). The Wellcome Centre for Cell Biology is supported by core funding from the Wellcome Trust (203149). H.-H.P. was sponsored by a PhD fellowship from the Chinese Scholarship Council. Y.-H.H. was sponsored by a PhD fellowship from the Chinese Scholarship Council and by a postdoctoral fellowship from the Collaborative Research Center 973 of the Deutsche Forschungsgemeinschaft.

## Author contributions

H.-H.P. cloned genes, produced proteins/complexes and performed experiments with help from Y.-H.H., Y.G., and N.S., prepared cryoEM samples with T.H., built atomic models with B.L. and M.C.W., refined structures with B.L. and performed crosslinking with Z.A.C. T.H., acquired, processed, and refined cryoEM data. Z.A.C. performed CLMS analyses. All authors contributed to the analysis of the data and the interpretation of the results. M.C.W., I.A., and G.B. wrote the manuscript with contributions from the other authors. J.R. and M.C.W. supervised work in their respective groups. M.C.W. conceived and coordinated the project.

## Funding

## Competing interests

The authors declare no competing interests.
