## [Peer Review File · Nature Communications]

Reviewer #1 (Remarks to the Author):

This manuscript reports cryo-EM structures of *Bacillus subtilis* RNA polymerase in complex with the RNAP recycling factor HelD. These structures, for the first time, provide a clear explanation of how an RNAP recycling factor interacts with RNAP and dislodges nucleic acids from it after transcription termination. Moreover, the structures also provide interaction details of RNAP with two accessory subunits--the delta and epsilon subunits that are only present in certain bacteria clades.

Transcription is a complex process tightly regulated in all domains of life. Although hundreds of structures of bacterial and eukaryotic RNAP were determined, very few RNAP structures were determined at the stage of transcription termination and post-termination. This work provides a significant advance of understanding post-termination regulation of bacterial transcription. The manuscript is well written, and the figures are clear. I highly recommend its publication in *Nature Communications*.

Two minor comments:

1. Previous reports suggested that the delta subunit of RNAP alone is able to promote RNAP recycling. The current structure shows that the delta-NTD docks on the RNAP and the D/E-rich delta-CTD interacts with HelD. Although it explains the synergistic effect of HelD and the delta subunit on RNAP recycling, it is still unknown how the delta subunit functions by itself, especially in some bacteria without the HelD orthologs. A brief discussion in the text would be more informative.

2. The RNAP concentration used for cryo-EM study is extremely high compared with the cellular physiological concentration, how would the author demonstrate the existence of the dimeric RNAP in physiological condition or in bacterial cells? As HelD is not directly involved in the dimerization interface, could the *Bacillus subtilis* RNAP dimerize by itself? How would the association of HelD affect RNAP dimerization? Is the dimerization state of RNA core enzyme compatible with sigma engagement? A more expanded discussion on this would be necessary.

Reviewer #2 (Remarks to the Author):

The structure of the *B. subtilis* RNAP-HelD- δ complex represents an important step forward in our understanding of the mechanism of RNAP recycling and the role it plays in transcription regulation in a subset of Gram-positive bacteria. It has been known for a while that HelD and δ factor together perform a complementary role in the stimulation of transcription in *B. subtilis* and that RNAP recycling is a key element of their biological role, but the molecular mechanism by which these factors brought about this stimulatory effect remained elusive. In this study, Wahl and colleagues have successfully reconstructed the RNAP-HelD- δ complex devoid of nucleic acid, essentially in the 'rescued' state, poised for σ factor exchange and RNAP recycling. They have clearly presented the role of the various δ factor and HelD domains in disrupting RNAP-nucleic acid interactions and imparting large conformational rearrangements which will promote the eviction of bound DNA and RNA from stalled RNAP complexes. They support this structural information with some important biochemical data, most important being the role of ATP binding in the final recycling step, the loss of HelD itself. As such, this work will make an important contribution to the field. However, in my opinion that the authors have made a misjudgment in the presentation and discussion of their results and placed too much emphasis (e.g. incorporated into the title) on a HelD-mediated RNAP

hibernation model which was not substantiated with data in the current study and therefore remains speculative. This detracts from important considerations (somewhat weak in the paper) of upstream events where HelD and δ factor must recognize and act upon stalled elongation complexes with stably associated nucleic acid. They make very little reference to early work which elegantly present the stimulatory role of HelD and δ factor in transcription assays and the differing effect of these factors on supercoiled plasmid and linear DNA fragments. This point is exemplified by their failure to address a central question that arises from earlier studies as well as their own work, namely, if δ factor alone is sufficient for efficient DNA eviction (RNAP recycling) then why is HelD, acting with δ factor, required to bring about the ~ 10 fold stimulation of transcription seen in transcription assays performed in earlier work? The paper would be much improved by shifting focus away from their interesting, yet still speculative hibernation model, towards central questions in the role of these factors in stimulating transcription.

Aside from the more general and overarching point raised above, the authors should also address the following list of concerns:

Major points:

1. The manuscript title should more closely represent the data presented, i.e. the functional state of the structure, rather than their more speculative hibernation model.
2. Wording in the abstract (pg. 2 lines 30-33) should be tempered to reflect the speculative and currently unsubstantiated status of their hibernation and nutritional sensing model.
3. Introductory paragraphs dealing with δ factor and HelD should include discussion of the stimulatory role of these factors in transcription assays and some of the mechanistic insights currently available in the literature. If word limits allow, it would be beneficial to conclude the introduction with a summary of the approaches used as well as the main findings and how they further our mechanistic understanding.
4. The authors fail to present direct evidence that HelD and δ factor displace pre-formed RNAP-nucleic acid scaffolds. Their preparation of material for cryo-EM appears to be their only attempt to pre-form the RNAP-nucleic acid (mismatch bubble) scaffold prior to incubation with HelD and δ factor. They state that HelD and δ factor displace this pre-bound nucleic acid, but provide no evidence that they have successfully formed the pre-requisite RNAP-nucleic acid complex. Such evidence should be presented, e.g. supplementary data. Extending this point, the authors repeatedly misrepresent their EMSA assay as a DNA displacement assay when in fact it is a direct competition assay as the factors are essentially all added together. The authors should address the following questions in regards to this assay, and ideally present novel data from repeats of this assay with appropriate modifications to the experimental parameters: 1) why was the assay performed in the absence of RNA to stabilize the mismatch bubble scaffold? 2) why was the assay performed in a buffer composed of 300mM monovalent salt rather than a physiological salt concentration, more appropriate for DNA-protein interactions (e.g. ~ 150 mM [monovalent ions]) 3) Why was the assay not performed with pre-formed RNAP-nucleic acid complexes to more appropriately represent nucleic acid 'displacement' rather than competition.

For the reasons stated above, the author's assertions that neither ATPase activity (pg. 8 line 190) nor DNA binding (pg. 8 line 196) are required for nucleic acid displacement by HelD and δ factor are not substantiated by the work as it stands and should not be included. As a matter of fact, they represent central mechanistic questions that remain unaddressed. Indeed, it seems unlikely that DNA binding, a function previously attributed to HelD sequences (outside of the N-terminus), does

not play an important role in the HeID-mediated nucleic acid displacement mechanism.

5. Wording regarding the suggested role of RNAP- δ factor-HeID dimerization in RNAP dormancy should be tempered throughout the text (e.g. pg. 9 lines 221-222, pg. 11 lines 258-270 and pg. 12 lines 292-297) in the absence of substantiating data. In their place, a description of potential future experiments to validate the model would be well placed, such as super-resolution live-cell imaging of *B. subtilis* harbouring fluorescent probes on HeID and RNAP at different growth stages.

6. Given the dramatic mode of HeID- δ factor C-term insertion into the RNAP DNA channel it was highly surprising to observe the apparent lack of effect on RNAP domain motion of the presence or absence of δ factor (similar #allowed/#disallowed β - β' crosslink ratios with/without δ factor in Fig. 4D&E). However, this surprising finding was not commented on in the manuscript but rather it was asserted that the δ factor N-term plays a role in promoting the open RNAP conformation. This discrepancy should be dealt with appropriately in the text and the full implications of the result suitably discussed.

Minor issues:

1. The authors should clarify the logic of the statement made regarding 'HeID retaining physiologically-relevant affinity' on pg. 9 lines 208-210.

2. Colour choices in some figures are poor and impact on clarity (e.g. use of white in Fig. 2 and Fig. 7)

3. The placement of ϵ subunit is not central to the key biological story and so the placement of Figure 2 detracts from the overall flow. I would suggest moving this to supplementary materials.

4. For clarity, I would suggest trying to maintain consistent molecular views where possible. For example, I see no good reason to show the molecules in Fig. 3C from a different viewing angle than Fig. 3A.

5. The data presented in Fig. 4D is slightly misleading and should be represented using a relative y-axis scale. There are two factors in play affecting the β - β' cross-linking observations, i) overall number of crosslinks, which varies between experiments, and ii) relative number of allowed (<25Å) and disallowed (>25Å) cross-links, which reflects the configuration of the main cleft. The allowed/disallowed β - β' crosslinking ratio should be clearly presenting by using a relative scale that removes the influence of the overall number of observed cross-links.

6. Extended data Fig. 6 shows that δ factor association with RNAP is not sensitive to nucleotide binding (in contrast to HeID). This result should be discussed in the manuscript and its consequences on RNAP recycling addressed.

Reviewer #3 (Remarks to the Author):

Thank you for sending the manuscript by Hao-Hong Pei et al. The present manuscript describes a complex between *Bacillus subtilis* RNA polymerase (RNAP) and HeID. HeID is a factor proposed to be involved in recycling RNAP stalled on DNA and unable to enter a new round of transcription. HeID may also actively contribute to shutting down transcription during phases of limiting nutrients.

The authors used single particle cryo-EM to obtain two reconstructions: a monomeric RNAP bound to a single copy of HeID and a dimeric RNAP bound to HeID, where the dimerization is driven by two RNAP monomers interacting with each other. The authors used cross-linking mass spectrometry and biochemistry to verify their structure and propose an attractive model for the role of HeID.

I think the present manuscript provides a number of important new insights, which will be

interesting for the transcription field and should be published once the authors have addressed the questions raised below.

Major comments:

The authors kindly provided maps and model but this reviewer noticed that the model (Monomeric-RNAP-HeID_D_1292109142_model-annotate_P1.pdb) does not fit either map (Monomeric-RNAP-HeID.mrc or Monomeric-RNAP-HeID_localfilter.mrc) but requires a shift (see attached screenshot) – was the model refined against this map? The authors should make sure maps and models agree before deposition.

On page 4, the authors argue that the conformational change of RNAP as a result of HeID binding causes clashes with the omega subunit and explains the loss of omega – could the authors explain this in more detail or provide a figure? In an alignment with the E. coli EC (6ALH) it appears that the omega binding site is less concave in the present structure. In other words, it looks to me it is less like a pocket or surface depression that accommodates and interacts with omega. The only possible clash is with residues in B. subtilis beta' (~620-640) but their repositioning seems more likely to be the result of omega loss rather than the cause.

On page 7 the authors describe unexplained density around HeIDBumper and model it as three consecutive helices of the DELTA-CTR. However, this reviewer thinks this is too speculative to be included in the coordinates for deposition because the provided maps are not of sufficient quality to support the model. The CTR portion should therefore not be deposited in my opinion.

On page 8 the authors describe experiments that suggest a cooperative mode of action for HeID and the RNAP delta subunit in releasing DNA from RNAP (or preventing binding in the first place). It sounds like these experiments were done with DNA only (containing an artificial bubble) and all components were mixed at the same time. I think it would be very informative to do additional experiments and:

- a) Pre-incubate RNAP alone with DNA prior to adding delta and/or HeID and compare to the present results; and
- b) Do the experiments also in presence of a complementary RNA transcript.

This would potentially provide information if an EC is less susceptible to Delta/HeID action and if the order of events matter.

Also on page 8, the authors describe the effect of ATP (analogues) on HeID binding – does HeID hydrolyze ATP in presence (or absence) of RNAP? If the presence of RNAP causes ATP hydrolysis, it would explain why more HeID is released in presence of AMPPNP or ATP-gamma-S.

On page 9, the authors briefly describe an RNAP dimer - can they exclude that the presence of the detergent caused RNAP dimerization – negative stain (and to a lesser extent SEC) +/- detergent would provide a quick answer.

On page 22, the authors describe data collection and processing. They mention data was collected at 0.832Å/pixel, 4x binned for initial processing. However, at the bottom of page 22, they state particles were re-extracted at a pixel size of 1.3Å/pixel (consistent with the provided maps) – why were the images scaled to a different pixel size? This is not good practice as it may introduce interpolation artifacts and it is not necessary in this case because only one dataset was collected (so no need to combine data from different microscopes).

It appears the same box size was used for mono- and dimers. The dimers have dimensions of 260Å. A boxsize of 440 pixels ($440 \times 0.832 \text{Å/pix} = 280 \times 1.307 \text{Å/pix} = \sim 366 \text{Å}$) is very small for the dimer and may result in loss of high-resolution information, which is displaced as a result of the CTF (see Henderson and Rosenthal JMB 2003). The extent of displacement of high-resolution information depends on the applied defocus – e.g. at 2.5µm defocus, the boxsize should be 1.5x bigger (i.e. ~ 620 pixels) to capture information to 4Å. The authors may consider a larger box size to gain more high-resolution information.

Minor comments:

On page 4, the authors note that RNAP purified from the Δ HeID strain had essentially no omega subunit but RNAP purified from Δ HeID Δ Delta, did have omega – that seems odd and the reviewer wonders if the loss of omega is specific to the delta HeID strain or just an effect of differences in the purification stringency (i.e. omega is weakly bound and tends to dissociate).

Related to the previous comment, Extended Data Fig. 1C shows an SEC run of RNAP supplemented with HeID and Delta and the authors argue that omega was underrepresented – however, from the gel we cannot judge because omega likely ran out of the gel. Could the authors provide a gel, where we can see (substoichiometric) omega.

The authors argue that the local resolution extends well beyond 3Å (Extended Data Fig. 3b). Judging from the maps, which were kindly provided, this reviewer thinks that it is likely overestimated and encourages the authors to use alternative software to independently confirm their estimates (e.g. blocres)

Response to reviewer comments

Revision summary

We thank the reviewers for their constructive comments, which we believe have helped to further strengthen our paper. In revising the manuscript, we have carefully taken into account all points raised by the reviewers, as detailed below. Briefly, we have conducted all additional experiments suggested by the reviewers (displacement assays with DNA and RNA, negative stain analyses, additional SEC/MALS analyses, updated gel analyses showing all relevant RNAP components). All new results are fully in line with our original interpretations. We present all these additional results in the revised text and in new figure panels. As suggested, we have also shifted the focus of our manuscript away from the hibernation model, which is also reflected in the new title. Relevant text changes and additions are highlighted. Coordinate files and maps are available to the reviewers under this updated link:

<https://box.fu-berlin.de/s/9DJtPo7ynAJg7gz>

In the point-by-point replies below, reviewer comments are repeated in bold italics, responses are in regular font, changed text passages are highlighted in yellow.

Reviewer #1

This manuscript reports cryo-EM structures of Bacillus subtilis RNA polymerase in complex with the RNAP recycling factor HeID. These structures, for the first time, provide a clear explanation of how an RNAP recycling factor interacts with RNAP and dislodges nucleic acids from it after transcription termination. Moreover, the structures also provide interaction details of RNAP with two accessory subunits--the delta and epsilon subunits that are only present in certain bacteria clades. Transcription is a complex process tightly regulated in all domains of life. Although hundreds of structures of bacterial and eukaryotic RNAP were determined, very few RNAP structures were determined at the stage of transcription termination and post-termination. This work provides a significant advance of understanding post-termination regulation of bacterial transcription. The manuscript is well written, and the figures are clear. I highly recommend its publication in Nature Communications.

We thank the reviewer for considering our work a significant advance and our manuscript to be well written, as well as for recommending publication in *Nature Communications*.

Two minor comments:

1. Previous reports suggested that the delta subunit of RNAP alone is able to promote RNAP recycling. The current structure shows that the delta-NTD docks on the RNAP and the D/E-rich delta-CTD interacts with HeID. Although it explains the synergistic effect of HeID and the delta subunit on RNAP recycling, it is still unknown how the delta subunit functions by itself, especially in some bacteria without the HeID orthologs. A brief discussion in the text would be more informative.

We thank the reviewer for this suggestion and have included a section on the possible mechanisms of δ during initiation and elongation in the revised manuscript:

Our cryoEM structures also inform about likely mechanisms of action of the δ subunit during initiation and elongation. Previously, δ alone had been shown to displace nucleic acids from RNAP⁷, a result we recapitulate here (Fig. 3g). As δ^{CTR} peptides showed similar activity when added in excess and as δ^{NTD} was found to bind RNAP, δ -mediated nucleic acid displacement was suggested to involve δ^{NTD} -dependent tethering of the polyanionic δ^{CTR} to core RNAP⁷. Our cryoEM structures confirm and further refine this hypothesis. δ^{NTD} anchors δ^{CTR} at the rim of the main channel; due to its length and intrinsic disorder, δ^{CTR} can reach into and explore most of the volume of the main channel, spatially and electrostatically competing with bound nucleic acids. This mode of action would explain how δ enhances core RNAP recycling in multi-round assays^{7,22}, and it may constitute the main recycling mechanism in bacteria that contain δ but lack HelD. It also provides an explanation for the finding that in vitro, RNAP- δ in the presence of σ factors still binds promoters and forms closed complexes, but fails to establish contacts with the downstream DNA^{22,49,50}, which are required for transition to an open complex. Finally, the model suggests that abolishing a positively charged region at the δ^{CTR} N-terminus, thus promoting more extended conformations of the CTR²³, effectively abrogates a restraint on δ^{CTR} 's ability to invade the main channel, thus reconciling increased effects of such CTR variants at promoters that form unstable complexes²³.

δ exhibits negative cooperativity with σ^{A} and favors its exchange for alternative σ factors that lack $\sigma 1.1$ ^{17,51}. In the *E. coli* σ^{70} holoenzyme, $\sigma 1.1$ can reside in the main channel, preventing access of either double- or single-stranded DNA to the RNAP active site³⁴ (Fig. 3d). To allow for DNA loading, the clamp has to open further⁵² or $\sigma 1.1$ has to move⁵³. These observations suggest binding competition between δ and $\sigma 1.1$, fully in line with our structures (Fig. 3c-e). However, while δ^{NTD} resembles the globular domain of $\sigma 1.1$ ³², our results indicate that the structurally unrelated CTR (together with HelD^{Bumper} if present) constitutes the $\sigma 1.1$ -competitive element that can occupy equivalent regions in the main channel (Fig. 3c).

We also have added more background information regarding δ to the introduction:

RNAPs from some Gram-positive bacteria, including *Bacillus subtilis*, contain additional small nonessential subunits, δ and ϵ . δ is present in *B. subtilis* at equal or higher concentration than standard core subunits, and its expression increases during the transition to the stationary phase^{16,17}, but δ deletion does not prevent sporulation^{17,18}. Cells lacking the *rpoE* gene, encoding δ , have altered morphology and exhibit an extended lag phase¹⁷ and defects in adaptation to changes in growth conditions sensed by initiating NTPs¹⁹. While a $\Delta rpoE$ strain has only mild phenotypes, it is not able to compete with the wild type (WT) strain¹⁹ and δ is required for virulence in *Streptococci*^{20,21}. δ destabilizes RNAP interactions with promoter DNA, inhibiting initiation at promoters that form unstable open complexes.^{19,22,23} Consequently, δ suppresses initiation from weak or cryptic promoters, and deletion of *rpoE* leads to expression of many otherwise silenced genes in *Streptococci*^{21,24}. Notably, δ also promotes RNAP recycling²² by displacing σ from holoenzyme²⁵ and RNA or DNA from binary complexes⁷. Presently, it is unclear how δ elicits these effects.

2. The RNAP concentration used for cryo-EM study is extremely high compared with the cellular physiological concentration, how would the author demonstrate the existence of the dimeric RNAP in physiological condition or in bacterial cells? As HelD is not directly involved in the dimerization interface, could the *Bacillus subtilis* RNAP dimerize by itself? How would the association of HelD affect RNAP dimerization? Is the dimerization state of RNA core enzyme compatible with sigma engagement? A more expanded discussion on this would be necessary.

Actually, estimating the nominal concentration of RNAP in log phase *B. subtilis* (from a combination of transcriptome and ribosome profiling data; about 25,000 copies of RNAP per cell; cell volume 1.41 μm^3), suggests that the RNAP concentration we use during our analyses is at least an order of magnitude lower. We acknowledge that the estimate of the in vivo concentration is only approximate and the actual in vivo activity may be strongly influenced by the effective water concentration in cells. However, this rough estimate shows that we are working most likely well below the actual in vivo concentration. The copy number of δ is equal to or higher than that of standard core subunits (as we now clearly state in the revised manuscript, see above). During sporulation, HeID concentration is at par with RNAP. We therefore think that (RNAP- δ -HeID)₂ can be easily envisaged to form in vivo.

We now also provide additional evidence for a fraction of RNAP- δ -HeID complexes forming dimers in solution (detected by SEC/MALS in the presence of 0.15 % n-octylglucoside as used for cryoEM analysis) and in negative stain EM both in the presence or absence of the detergent (please refer to new Supplementary Figure 1d,e).

We now briefly mention these considerations and new results in the revised manuscript:

About two thirds of our cryoEM particle images conformed to dimeric (RNAP- δ -HeID)₂ complexes (Fig. 6c; Supplementary Note 3), which were partially stable during SEC under conditions identical to cryoEM sample preparation (0.15 % n-octylglucoside; Supplementary Figure 1d). We also conducted negative stain EM analyses with RNAP- δ -HeID in the presence or absence of 0.15 % n-octylglucoside and detected dimers under both conditions (Supplementary Figure 1e; a quantitative analysis of the monomer/dimer distribution was precluded by preferred particle orientations on the carbon films).

...

Intriguingly, we observed (RNAP- δ -HeID)₂ dimers resembling hibernating eukaryotic RNAP I (Fig. 6c,d), which were partially stable in SEC at initial RNAP concentrations about 10-fold lower compared to their nominal cellular concentrations in the log phase, estimated from transcript levels and ribosome profiling^{30,55}.

Supplementary Figure 1: Complex preparations.

d, SEC/multi-angle light scattering analysis of RNAP- δ -HelD used for cryoEM analysis in buffer lacking (-OG) or containing (+OG) 0.15 % (w/v) n-octylglucoside (critical micellar concentration 0.6 % [w/v]). Black traces, UV signals; red lines, molecular mass estimates across the peaks. Molecular masses deduced are listed in the bottom table compared to the theoretical (theor.) molecular masses for RNAP- δ -HelD (Mon.) and (RNAP- δ -HelD)₂ (Dimer). About 16 % of the sample traverses the column as intact (RNAP- δ -HelD)₂ dimers in the presence of n-octylglucoside.

e, Top, negative stain EM micrographs of RNAP- δ -HelD in buffer lacking (-OG) or containing (+OG) 0.15 % (w/v) n-octylglucoside. Scale bars, 100 nm. Bottom, 2D class averages of picked particle images. Classes boxed red unequivocally indicate the presence of (RNAP- δ -HelD)₂ dimers in both samples.

The negative stain EM analysis is described in the revised Methods:

Negative stain EM analysis

RNAP- δ -HelD complex was prepared as for cryoEM analysis, diluted to 25 $\mu\text{g/ml}$ in buffer H and supplemented with 0.15 % n-octylglucoside or buffer immediately before grid preparation. 4 μl of the samples were added to glow-discharged Formvar/carbon grids (S162, Plano GmbH), left to settle for 40 s and manually blotted with Whatman paper No. 1, followed by addition of 4 μl of 1 % (w/v) uranyl acetate staining solution. After 40 s incubation, the grids were manually blotted and dried at ambient temperature overnight. Samples were imaged on an FEI Talos L120C TEM, operated at 120 kV, equipped with an FEI CETA 16M CCD camera at a nominal magnification of 57,000x. The calibrated pixel size was 1.96 $\text{\AA}/\text{px}$. Images were acquired manually in low dose mode using TEM Imaging & Analysis (TIA) software, supplied by the manufacturer, accumulating a total electron dose of 50 $\text{e}^-/\text{\AA}^2$. Image analysis was done with cryoSPARC. After CTF estimation, manually selected particle images were used as reference for template-based particle picking. Particle images were extracted with a box size of 160 px and resampled to 80 px. A mask of 220 \AA diameter was applied during 2D classification.

Reviewer #2

The structure of the B. subtilis RNAP-HelD- δ complex represents an important step forward in our understanding of the mechanism of RNAP recycling and the role it plays in transcription regulation in a subset of Gram-positive bacteria. It has been known for a while that HelD and δ factor together perform a complementary role in the stimulation of transcription in B. subtilis and that RNAP recycling is a key element of their biological role, but the molecular mechanism by which these factors brought about this stimulatory effect remained elusive. In this study, Wahl and colleagues have successfully reconstructed the RNAP-HelD- δ complex devoid of nucleic acid, essentially in the 'rescued' state, poised for σ factor exchange and RNAP recycling. They have clearly presented the role of the various δ factor and HelD domains in disrupting RNAP-nucleic acid interactions and imparting large conformational rearrangements which will promote the eviction of bound DNA and RNA from stalled RNAP complexes. They support this structural information with some important biochemical data, most important being the role of ATP binding in the final recycling step, the loss of HelD itself. As such, this work will make an important contribution to the field.

We thank the reviewer for considering our work an important step forward and an important contribution to the field.

However, in my opinion that the authors have made a misjudgment in the presentation and discussion of their results and placed too much emphasis (e.g. incorporated into the title) on a HelD-mediated RNAP hibernation model which was not substantiated with data in the current study and therefore remains speculative. This detracts from important considerations (somewhat weak in the paper) of upstream events where HelD and δ factor must recognize and act upon stalled elongation complexes with stably associated nucleic acid. They make very little reference to early work which elegantly present the stimulatory role of HelD and δ factor in transcription assays and the differing effect of these factors on supercoiled plasmid and linear DNA fragments.

We agree with the reviewer and have shifted our focus away from the hibernation model and towards functions of δ and HelD during elongation and recycling. We apologize for not

referencing some relevant previous work, in part due to space limitations. While we did not find much additional earlier work on HeID than what we referred to already in the first version of our manuscript, we now refer to additional earlier work on δ in the revised manuscript:

RNAPs from some Gram-positive bacteria, including *Bacillus subtilis*, contain additional small nonessential subunits, δ and ϵ . δ is present in *B. subtilis* at equal or higher concentration than standard core subunits, and its expression increases during the transition to the stationary phase^{16,17}, but δ deletion does not prevent sporulation^{17,18}. Cells lacking the *rpoE* gene, encoding δ , have altered morphology and exhibit an extended lag phase¹⁷ and defects in adaptation to changes in growth conditions sensed by initiating NTPs¹⁹. While a $\Delta rpoE$ strain has only mild phenotypes, it is not able to compete with the wild type (WT) strain¹⁹ and δ is required for virulence in *Streptococci*^{20,21}. δ destabilizes RNAP interactions with promoter DNA, inhibiting initiation at promoters that form unstable open complexes.^{19,22,23} Consequently, δ suppresses initiation from weak or cryptic promoters, and deletion of *rpoE* leads to expression of many otherwise silenced genes in *Streptococci*^{21,24}. Notably, δ also promotes RNAP recycling²² by displacing σ from holoenzyme²⁵ and RNA or DNA from binary complexes⁷. Presently, it is unclear how δ elicits these effects.

This point is exemplified by their failure to address a central question that arises from earlier studies as well as their own work, namely, if δ factor alone is sufficient for efficient DNA eviction (RNAP recycling) then why is HeID, acting with δ factor, required to bring about the ~10x fold stimulation of transcription seen in transcription assays performed in earlier work? The paper would be much improved by shifting focus away from their interesting, yet still speculative hibernation model, towards central questions in the role of these factors in stimulating transcription.

We appreciate the expert opinion of the reviewer and have shifted our focus in the revised manuscript as suggested (see also previous and following points). E.g., we now discuss implications of our findings on the function of δ alone during initiation and elongation in much more depth in the revised version:

Our cryoEM structures also inform about likely mechanisms of action of the δ subunit during initiation and elongation. Previously, δ alone had been shown to displace nucleic acids from RNAP⁷, a result we recapitulate here (Fig. 3g). As δ^{CTR} peptides showed similar activity when added in excess and as δ^{NTD} was found to bind RNAP, δ -mediated nucleic acid displacement was suggested to involve δ^{NTD} -dependent tethering of the polyanionic δ^{CTR} to core RNAP⁷. Our cryoEM structures confirm and further refine this hypothesis. δ^{NTD} anchors δ^{CTR} at the rim of the main channel; due to its length and intrinsic disorder, δ^{CTR} can reach into and explore most of the volume of the main channel, spatially and electrostatically competing with bound nucleic acids. This mode of action would explain how δ enhances core RNAP recycling in multi-round assays^{7,22}, and it may constitute the main recycling mechanism in bacteria that contain δ but lack HeID. It also provides an explanation for the finding that in vitro, RNAP- δ in the presence of σ factors still binds promoters and forms closed complexes, but fails to establish contacts with the downstream DNA^{22,49,50}, which are required for transition to an open complex. Finally, the model suggests that abolishing a positively charged region at the δ^{CTR} N-terminus, thus promoting more extended conformations of the CTR²³, effectively abrogates a restraint on δ^{CTR} 's ability to invade the main channel, thus reconciling increased effects of such CTR variants at promoters that form unstable complexes²³.

δ exhibits negative cooperativity with σ^{A} and favors its exchange for alternative σ factors that lack $\sigma 1.1$ ^{17,51}. In the *E. coli* σ^{70} holoenzyme, $\sigma 1.1$ can reside in the main channel, preventing

access of either double- or single-stranded DNA to the RNAP active site³⁴ (Fig. 3d). To allow for DNA loading, the clamp has to open further⁵² or σ 1.1 has to move⁵³. These observations suggest binding competition between δ and σ 1.1, fully in line with our structures (Fig. 3c-e). However, while δ^{NTD} resembles the globular domain of σ 1.1³², our results indicate that the structurally unrelated CTR (together with HeID^{Bumper} if present) constitutes the σ 1.1-competitive element that can occupy equivalent regions in the main channel (Fig. 3c).

Aside from the more general and overarching point raised above, the authors should also address the following list of concerns:

Major points:

1. The manuscript title should more closely represent the data presented, i.e. the functional state of the structure, rather than their more speculative hibernation model.

We agree and have adjusted the title to:

The δ subunit and NTPase HeID institute a two-pronged mechanism for RNA polymerase recycling

2. Wording in the abstract (pg. 2 lines 30-33) should be tempered to reflect the speculative and currently unsubstantiated status of their hibernation and nutritional sensing model.

We agree and have adjusted the Abstract according to the reviewer's suggestions. A single sentence relating to dimeric (RNAP- δ -HeID)₂ complexes now reads:

HeID abundance during slow growth and a dimeric (RNAP- δ -HeID)₂ structure that resembles hibernating eukaryotic RNAP I suggest that HeID might also modulates active enzyme pools in response to cellular cues.

3. Introductory paragraphs dealing with δ factor and HeID should include discussion of the stimulatory role of these factors in transcription assays and some of the mechanistic insights currently available in the literature. If word limits allow, it would be beneficial to conclude the introduction with a summary of the approaches used as well as the main findings and how they further our mechanistic understanding.

While we did not find much additional earlier work on HeID than what we referred to already in the first version of our manuscript, we now refer to additional earlier work on δ in the revised manuscript:

RNAPs from some Gram-positive bacteria, including *Bacillus subtilis*, contain additional small nonessential subunits, δ and ϵ . δ is present in *B. subtilis* at equal or higher concentration than standard core subunits, and its expression increases during the transition to the stationary phase^{16,17}, but δ deletion does not prevent sporulation^{17,18}. Cells lacking the *rpoE* gene, encoding δ , have altered morphology and exhibit an extended lag phase¹⁷ and defects in adaptation to changes in growth conditions sensed by initiating NTPs¹⁹. While a $\Delta rpoE$ strain has only mild phenotypes, it is not able to compete with the wild type (WT) strain¹⁹ and δ is

required for virulence in *Streptococcus*^{20,21}. δ destabilizes RNAP interactions with promoter DNA, inhibiting initiation at promoters that form unstable open complexes.^{19,22,23} Consequently, δ suppresses initiation from weak or cryptic promoters, and deletion of *rpoE* leads to expression of many otherwise silenced genes in *Streptococcus*^{21,24}. Notably, δ also promotes RNAP recycling²² by displacing σ from holoenzyme²⁵ and RNA or DNA from binary complexes⁷. Presently, it is unclear how δ elicits these effects.

We also now conclude the Introduction with a short summary section:

Using single-particle cryogenic electron microscopy (cryoEM) and crosslinking/mass spectrometry (CLMS), we show that HelD, supported by δ , inserts long prongs into RNAP's main and secondary channels, competing with bound nucleic acids and prying RNAP open to allow nucleic acid escape. Release assays further support HelD/ δ collaboration in RNAP recycling. ATP facilitates HelD detachment and completes RNAP recovery. We also observe RNAP dimerization in the presence of δ and HelD, hinting at a possible role of HelD in RNAP hibernation.

4. The authors fail to present direct evidence that HelD and δ factor displace pre-formed RNAP-nucleic acid scaffolds. Their preparation of material for cryo-EM appears to be their only attempt to pre-form the RNAP-nucleic acid (mismatch bubble) scaffold prior to incubation with HelD and δ factor. They state that HelD and δ factor displace this pre-bound nucleic acid, but provide no evidence that they have successfully formed the prerequisite RNAP-nucleic acid complex. Such evidence should be presented, e.g. supplementary data.

Extending this point, the authors repeatedly misrepresent their EMSA assay as a DNA displacement assay when in fact it is a direct competition assay as the factors are essentially all added together. The authors should address the following questions in regards to this assay, and ideally present novel data from repeats of this assay with appropriate modifications to the experimental parameters: 1) why was the assay performed in the absence of RNA to stabilize the mismatch bubble scaffold? 2) why was the assay performed in a buffer composed of 300mM monovalent salt rather than a physiological salt concentration, more appropriate for DNA-protein interactions (e.g. ~150mM [monovalent ions]) 3) Why was the assay not performed with pre-formed RNAP-nucleic acid complexes to more appropriately represent nucleic acid 'displacement' rather than competition.

First, our sincere apologies – we have not properly presented the experimental setup for our δ /HelD/DNA competitive binding assays. In these assays, we indeed first formed RNAP-DNA complexes (formation of these complexes can be seen from the band shifts with RNAP in the absence of δ and HelD). δ and/or HelD were only added later. Thus, the assays indeed represent DNA displacement assays. We have corrected the description in the Methods section:

Nucleic acid displacement assays

Equimolar amounts of 5'-[³²P]-labeled ntDNA and unlabeled tDNA capable of forming an artificial bubble, or additionally an RNA 9-mer with complementarity to the tDNA in the bubble (Supplementary Table 1), were mixed in buffer G and annealed by heating to 95 °C for 5 min and subsequent cooling to 25 °C at 1 °C/min. 20 nM (to achieve a 10 nM concentration in the final mix) of the labeled DNA duplex or DNA/RNA scaffold were incubated with 2 μ M (to achieve a 1 μ M concentration in the final mix) RNAP ^{$\Delta\delta\Delta$ HelD} in buffer G for 10 min at 4 °C, followed by an additional 10 min incubation at 32 °C. Subsequently, (i) buffer, (ii) HelD (1 μ M final

concentration); (iii) δ (1 μ M final concentration), (iv) combinations of HeID (1 μ M final concentration) and δ (titrated final concentration; see Fig. 3g) or (v) HeID and δ^{NTD} (1 μ M final concentration each) were added, and the samples were further incubated for 10 min at 32°C. Samples were loaded on a 4 % native PAGE gel and electrophoresed in 0.5X TBE buffer. Radioactive bands were visualized using a Storm phosphorimager and quantified using ImageQuant software (GE Healthcare).

For the original manuscript, we performed this assay with DNA harboring an artificial transcription bubble. We chose this DNA because it is expected to associate with RNAP in a manner that closely resembles a post-termination RNAP that remained stuck on the template (Ref. 40). Several recent papers show that RNAP frequently remains attached to DNA after termination (Ref. 3-5). Notably, such DNA-trapped RNAPs can initiate transcription in reverse direction (Ref. 5), which would pose a major threat to genome integrity due to head-on collisions with replisomes. Thus, we are convinced that our initial experiment monitors a highly relevant scenario of δ /HeID-mediated RNAP recycling. We now briefly present these arguments in the revised manuscript:

To further delineate the contributions of δ and HeID to nucleic acid displacement, we conducted band shift assays, in which we first bound RNAP to nucleic acids and subsequently added δ and/or HeID. We first tested displacement of DNA with an artificial bubble, which when bound to RNAP mimics a situation ensuing after many intrinsic termination events^{3-5,40}.

However, we fully agree that it is also important to monitor δ /HeID-mediated displacement of nucleic acids from an EC that contains DNA with a bubble stabilized by RNA, to assess whether the factors in principle can recycle stalled RNAPs. We therefore now also conducted these additional displacement assays. For analysis of EC disruption, we assembled ECs on a nucleic acid scaffold bearing an artificial bubble. As the DNA region forming the bubble cannot reanneal during displacement, EC disruption represents a particularly demanding task in our setup. We find that δ and HeID still quite efficiently displace the nucleic acids from such an EC, albeit less efficiently than DNA alone, as expected. We describe these additional results in the revised text and present them in revised Fig. 3g:

Next, we tested the ability of δ /HeID to dissociate ECs assembled on an artificial DNA bubble and complementary RNA, mimicking stalled ECs. A similar picture as for DNA-only displacement emerged; however, due to the RNA-mediated stabilization of DNA on RNAP, HeID and δ individually or HeID/ δ^{NTD} liberated less RNAP, and higher concentrations of δ in the presence of HeID were required to achieve full nucleic acid displacement (Fig. 3g, lanes 15-28).

Fig. 3: HeID/ δ -mediated RNAP recycling.

g, EMSA monitoring displacement of DNA (lanes 1-14) or DNA/RNA (lanes 15-28) from RNAP by HeID, δ or combinations. Top scheme, samples analyzed; gray boxes, respective component added (proteins in equimolar amounts to RNAP ^{$\Delta\delta\Delta$ HeID}). Numbers, molar ratios of δ or δ^{NTD} relative to RNAP ^{$\Delta\delta\Delta$ HeID} added. Panels labeled “DNA” or “DNA/RNA”, native PAGE analyses. All lanes are from the same gel, some lanes for the DNA-only gel were removed for display purposes (dashed line). Bar graphs, quantification of the data shown in the middle panels. Values represent means of DNA bound relative to RNAP ^{$\Delta\delta\Delta$ HeID} alone \pm SD for three independent experiments, using the same biochemical samples (data points indicated).

Regarding the salt concentration, we used 20 mM TRIS-HOAc, 5 mM Mg[OAc]₂, 100 mM KOAc, 2 mM DTT, 5 % [v/v] glycerol, pH 8.0 for displacement assays. For some other assays, much higher concentrations of RNAP were needed, and in our hands *B. subtilis* RNAP solubility strongly decreases upon reducing the salt concentration below 300 mM. We note that 300 mM salt is only about 2-fold above the physiological salt concentration and ECs are exceedingly stable even in 1 M salt. Therefore, we considered the salt concentrations we used in our buffers an acceptable compromise.

For the reasons stated above, the author's assertions that neither ATPase activity (pg. 8 line 190) nor DNA binding (pg. 8 line 196) are required for nucleic acid displacement by HeID and δ factor are not substantiated by the work as it stands and should not be included. As a matter of fact, they represent central mechanistic questions that remain unaddressed. Indeed, it seems unlikely that DNA binding, a function previously attributed to HeID sequences (outside of the N-terminus), does not play an important role in the HeID-mediated nucleic acid displacement mechanism.

We beg to disagree that our notion that ATPase activity is not required for nucleic acid displacement is unsubstantiated – we did not add ATP to our displacement assays and see efficient displacement. We thus kept a brief statement to this effect in the revised manuscript:

Notably, δ /HeID-mediated DNA or DNA/RNA displacement did not require addition of ATP.

However, while we do not see DNA or DNA/RNA binding by HeID in EMSA (Fig. 3g, lanes 2 and 16) and while our structures suggest that HeID-mediated RNAP recycling may not require such activity, we agree with the reviewer that our results do not disprove this possibility. We have therefore omitted the corresponding statement from the revised manuscript. We also have removed the original Fig. 3d, which we had included to support this idea.

5. Wording regarding the suggested role of RNAP- δ factor-HeID dimerization in RNAP dormancy should be tempered throughout the text (e.g. pg. 9 lines 221-222, pg. 11 lines 258-270 and pg. 12 lines 292-297) in the absence of substantiating data. In their place, a description of potential future experiments to validate the model would be well placed, such as super-resolution live-cell imaging of *B. subtilis* harbouring fluorescent probes on HeID and RNAP at different growth stages.

We agree and, as suggested, we have toned down our original suggestions on a role of (RNAP- δ -HeID)₂ dimers in RNAP dormancy. We would like to point out that, based on suggestions by Reviewer 3, we added some additional evidence for dimer formation in the revised manuscript, now presented in new Supplementary Figure 1d,e:

About two thirds of our cryoEM particle images conformed to dimeric (RNAP- δ -HeID)₂ complexes (Fig. 6c; Supplementary Note 3), which were partially stable during SEC under conditions identical to cryoEM sample preparation (0.15 % n-octylglucoside; Supplementary Figure 1d). We also conducted negative stain EM analyses with RNAP- δ -HeID in the presence or absence of 0.15 % n-octylglucoside and detected dimers under both conditions (Supplementary Figure 1e; a quantitative analysis of the monomer/dimer distribution was precluded by preferred particle orientations on the carbon films).

...

Intriguingly, we observed $(\text{RNAP-}\delta\text{-HeID})_2$ dimers resembling hibernating eukaryotic RNAP I (Fig. 6c,d), which were partially stable in SEC at initial RNAP concentrations about 10-fold lower compared to their nominal cellular concentrations in the log phase, estimated from transcript levels and ribosome profiling^{30,55}.

Supplementary Figure 1: Complex preparations.

d, SEC/multi-angle light scattering analysis of RNAP- δ -HeID used for cryoEM analysis in buffer lacking (-OG) or containing (+OG) 0.15 % (w/v) n-octylglucoside (critical micellar concentration 0.6 % [w/v]). Black traces, UV signals; red lines, molecular mass estimates across the peaks. Molecular masses deduced are listed in the bottom table compared to the theoretical (theor.) molecular masses for RNAP- δ -HeID (Mon.) and $(\text{RNAP-}\delta\text{-HeID})_2$ (Dimer). About 16 % of the sample traverses the column as intact $(\text{RNAP-}\delta\text{-HeID})_2$ dimers in the presence of n-octylglucoside.

e, Top, negative stain EM micrographs of RNAP- δ -HeID in buffer lacking (-OG) or containing (+OG) 0.15 % (w/v) n-octylglucoside. Scale bars, 100 nm. Bottom, 2D class averages of picked

particle images. Classes boxed red unequivocally indicate the presence of (RNAP- δ -HeID)₂ dimers in both samples.

Thus, we kept a brief discussion on the possible role of these dimers:

Intriguingly, we observed (RNAP- δ -HeID)₂ dimers resembling hibernating eukaryotic RNAP I (Fig. 6c,d), which were partially stable in SEC at initial RNAP concentrations about 10-fold lower compared to their nominal cellular concentrations in the log phase, estimated from transcript levels and ribosome profiling^{30,55}. Dimerization of RNAP has also been reported in bacteria that lack HeID, including *E. coli*⁹. While dimerization may thus be an inherent property of RNAPs, we observe a dimerization mode in the presence of HeID that involves homologous interactions between the β' clamps, the C-terminal β clamp and regions of the β flap; clearly, the concomitant usage of these contact points is facilitated by HeID pushing the β' clamp outwards (Fig. 6c). Thus, while not directly involved in forming the dimer interface, HeID clearly facilitates the observed mode of dimerization. Notably, comparison of our dimeric structure to a *M. smegmatis* RNAP- σ^A holoenzyme structure³⁸ shows that all binding sites for σ , except for $\sigma 1.1$ in the main channel, would be accessible in the RNAP- δ -HeID dimer. Thus, re-binding of σ could contribute to efficient recovery of RNAP from the dimeric state. Taken together, HeID/ δ could in principle promote RNAP hibernation that may be essential for fast RNAP recovery, in line with observations that overexpression of HeID enhances sporulation⁵⁶ and deletions of HeID, δ or both prolong the lag phase²⁹.

In addition, we included a brief statement in the Discussion on how the hibernation question may be addressed in the future, as suggested:

Further tests of this idea are required and could involve in vivo CLMS at different growth phases and during sporulation, in WT compared to $\Delta heID$ or overexpressing cells, or in vivo super-resolution imaging with fluorescence labeling of HeID or RNAP.

6. Given the dramatic mode of HeID- δ factor C-term insertion into the RNAP DNA channel it was highly surprising to observe the apparent lack of effect on RNAP domain motion of the presence or absence of δ factor (similar #allowed/#disallowed β - β' crosslink ratios with/without δ factor in Fig. 4D&E). However, this surprising finding was not commented on in the manuscript but rather it was asserted that the δ factor N-term plays a role in promoting the open RNAP conformation. This discrepancy should be dealt with appropriately in the text and the full implications of the result suitably discussed.

We apologize again for having been too terse in describing a presumed supporting role of the δ NTD in RNAP conformational changes. Compared to the conformational changes induced by HeID binding, the δ NTD effects are minor and would not be expected to be clearly reflected in CLMS data of RNAP +/- δ . Indeed, the overall conformation of our RNAP- δ -HeID complex is quite similar to the conformation of RNAP-HeID in the absence of δ as presented in an accompanying manuscript by another group. We now clarify this situation in the revised manuscript:

δ consists of a folded N-terminal domain (NTD; residues 1-90) and an intrinsically disordered acidic C-terminal region (CTR; residues 91-173) with a net -47 negative charge^{7,23}. As noted previously³², the first ~70 residues of δ^{NTD} resemble the globular domain of $\sigma 1.1$ regions of group 1 σ factors³³. However, unlike the $\sigma 1.1$ domain in an *E. coli* σ^{70} holoenzyme³⁴, δ^{NTD} does not reside in the main channel but binds on the surface of RNAP between the β' shelf and jaw

(Fig. 1a; Supplementary Figure 5a), in agreement with a previous *in vivo* CLMS analysis³⁵. Comparison to the *E. coli* EC³¹ showed that δ^{NTD} seems to contribute to main channel opening by somewhat contracting the jaw and β' shelf; furthermore, RNAP opening and slight δ^{NTD} -mediated displacement of the shelf lead to repositioning of β' secondary channel elements, which would clash with ω at its canonical binding site (Supplementary Figure 5a), explaining loss of ω in RNAP- δ and RNAP- δ -HeID complexes (Supplementary Figure 1b,c). Lack of continuous cryoEM density beyond δ^{NTD} shows that δ^{CTR} is suspended from the rim of the main channel in a flexible manner (see below).

We also included another Figure (new Supplementary Figure 5a), in which we illustrate conformational changes to which δ^{NTD} contributes:

Supplementary Figure 5: HeID- δ / ω competition and ϵ subunit.

a, Close-up view on δ^{NTD} and ω binding regions with ω transferred from an *E. coli* EC (PDB ID 6ALH) by superpositioning of the β subunits. δ^{NTD} and HeID displace the β' shelf and other secondary channel elements (black arrow), leading to steric hindrance of ω binding (red arrows).

Minor issues:

1. The authors should clarify the logic of the statement made regarding 'HeID retaining physiologically-relevant affinity' on pg. 9 lines 208-210.

We omitted the statement from the revised manuscript as it was not required.

2. Colour choices in some figures are poor and impact on clarity (e.g. use of white in Fig. 2 and Fig. 7).

We employed coloring to highlight specific regions or factors that are relevant to our work, such as the β' clamp, δ , ϵ , HelD and nucleic acids. Other regions of RNAP we would like to display in different shades of gray to not lose contrast and to not distract from the key features. To still distinguish different RNAP subunits, we chose as wide a grayscale as possible. We therefore would prefer to keep colors as originally suggested.

3. The placement of ϵ subunit is not central to the key biological story and so the placement of Figure 2 detracts from the overall flow. I would suggest moving this to supplementary materials.

As suggested, we moved original Fig. 2 to the supplement (now Supplementary Figure 5b).

4. For clarity, I would suggest trying to maintain consistent molecular views where possible. For example, I see no good reason to show the molecules in Fig. 3C from a different viewing angle than Fig. 3A.

We agree and have rearranged the view in original Fig. 3c to match that of original Fig 3a (now Fig. 2a,c; reduced version below).

5. The data presented in Fig. 4D is slightly misleading and should be represented using a relative y-axis scale. There are two factors in play affecting the β - β' cross-linking observations, i) overall number of crosslinks, which varies between experiments, and ii) relative number of allowed (<25Å) and disallowed (>25Å) cross-links, which reflects the configuration of the main cleft. The allowed/disallowed β - β' crosslinking ratio should

be clearly presenting by using a relative scale that removes the influence of the overall number of observed cross-links.

We believe that both the absolute numbers of crosslinks (matching and over-length) as well as their relative fractions are important pieces of information. We now provide both in the revised Fig. 4d:

Fig. 4: Structure probing by CLMS.

d, Numbers of crosslinks (bars) between β and β' identified from the four crosslinked complexes, and fractions of over-length crosslinks (percentages at the bottom). Crosslinks are color-coded as in (b). In the RNAP^{ΔδΔHeID}-δ-HeID complex, a significantly reduced number of β - β' over-length crosslinks (in reference to the RNAP-δ-HeID structure) compared to the RNAP^{ΔδΔHeID}, RNAP^{ΔδΔHeID}-δ and RNAP^{ΔδΔHeID}-HeID complexes suggests that δ and HeID cooperate to stabilize an open conformation of RNAP.

In addition, we now clarify in the text:

RNAP^{ΔδΔHeID}, RNAP^{ΔδΔHeID}-δ and RNAP^{ΔδΔHeID}-HeID yielded significantly more crosslinks than RNAP^{ΔδΔHeID}-δ-HeID and, among those, in particular many more over-length crosslinks when compared to the RNAP-δ-HeID structure (Fig. 4c,d). Furthermore, the fraction of crosslinks corresponding to over-length crosslinks was strongly increased in RNAP^{ΔδΔHeID} and RNAP^{ΔδΔHeID}-δ compared to complexes containing HeID (Fig. 4c,d). The reduced total number of crosslinks suggests a reduction in conformations explored by RNAP upon δ or HeID binding, and in particular when both factors are present. The reduced total number and fraction of over-length crosslinks suggests a conformation closer to our RNAP-δ-HeID cryoEM structure in the presence of HeID.

6. Extended data Fig. 6 shows that δ factor association with RNAP is not sensitive to nucleotide binding (in contrast to HeID). This result should be discussed in the manuscript and its consequences on RNAP recycling addressed.

We thank the reviewer for pointing this out and have included a brief description and discussion of this aspect in the revised manuscript:

Unlike HeID, δ is not displaced from RNAP by addition of ATP or analogs (Fig. 6b; Supplementary Figure 7b).

...

In contrast, ATP does not induce concomitant release of δ (Fig. 6b; Supplementary Figure 7b), confirming that δ has intrinsically high affinity for RNAP and does not require HeID to remain stably associated. As association of alternative σ factors (relative to σ^A) is favored in RNAP- δ compared to RNAP lacking $\delta^{17,51}$, additional mechanisms may be at play to remove δ (or expunge δ^{CTR} from the main channel) in situations where efficient rebinding of σ^A is specifically required.

Reviewer #3

Thank you for sending the manuscript by Hao-Hong Pei et al. The present manuscript describes a complex between Bacillus subtilis RNA polymerase (RNAP) and HeID. HeID is a factor proposed to be involved in recycling RNAP stalled on DNA and unable to enter a new round of transcription. HeID may also actively contribute to shutting down transcription during phases of limiting nutrients.

The authors used single particle cryo-EM to obtain two reconstructions: a monomeric RNAP bound to a single copy of HeID and a dimeric RNAP bound to HeID, where the dimerization is driven by two RNAP monomers interacting with each other. The authors used cross-linking mass spectrometry and biochemistry to verify their structure and propose an attractive model for the role of HeID.

I think the present manuscript provides a number of important new insights, which will be interesting for the transcription field and should be published once the authors have addressed the questions raised below.

We thank the reviewer for the positive overall evaluation and for considering our findings important and of interest to the transcription field.

Major comments:

The authors kindly provided maps and model but this reviewer noticed that the model (Monomeric-RNAP-HeID_D_1292109142_model-annotate_P1.pdb) does not fit either map (Monomeric-RNAP-HeID.mrc or Monomeric-RNAP-HeID_localfilter.mrc) but requires a shift (see attached screenshot) – was the model refined against this map? The authors should make sure maps and models agree before deposition.

We thank the reviewer for pointing out the disagreement between the monomeric RNAP- δ -HeID model and cryoEM densities we provided for review. We apologize, this was simply a mistake in mixing up files on our side. The model has indeed been refined against the maps deposited in the EMDB. The coordinates submitted to the PDB and the MRC files submitted to the EMDB are congruent. We are sorry for the confusion this must have caused and now provide the correct files for the reviewer's inspection under this link:

<https://box.fu-berlin.de/s/9DJtPo7ynAJq7gz>

On page 4, the authors argue that the conformational change of RNAP as a result of HeID binding causes clashes with the omega subunit and explains the loss of omega – could the authors explain this in more detail or provide a figure? In an alignment with the *E. coli* EC (6ALH) it appears that the omega binding site is less concave in the present structure. In other words, it looks to me it is less like a pocket or surface depression that accommodates and interacts with omega. The only possible clash is with residues in *B. subtilis* beta' (~620-640) but their repositioning seems more likely to be the result of omega loss rather than the cause.

As suggested, we further explained conformational changes that lead to displacement of ω and prepared an additional figure (new Supplementary Figure 5a) to illustrate the situation:

δ consists of a folded N-terminal domain (NTD; residues 1-90) and an intrinsically disordered acidic C-terminal region (CTR; residues 91-173) with a net -47 negative charge^{7,23}. As noted previously³², the first ~70 residues of δ^{NTD} resemble the globular domain of $\sigma 1.1$ regions of group 1 σ factors³³. However, unlike the $\sigma 1.1$ domain in an *E. coli* $\sigma 70$ holoenzyme³⁴, δ^{NTD} does not reside in the main channel but binds on the surface of RNAP between the β' shelf and jaw (Fig. 1a; Supplementary Figure 5a), in agreement with a previous *in vivo* CLMS analysis³⁵. Comparison to the *E. coli* EC³¹ showed that δ^{NTD} seems to contribute to main channel opening by somewhat contracting the jaw and β' shelf; furthermore, RNAP opening and slight δ^{NTD} -mediated displacement of the shelf lead to repositioning of β' secondary channel elements, which would clash with ω at its canonical binding site (Supplementary Figure 5a), explaining loss of ω in RNAP- δ and RNAP- δ -HeID complexes (Supplementary Figure 1b,c). Lack of continuous cryoEM density beyond δ^{NTD} shows that δ^{CTR} is suspended from the rim of the main channel in a flexible manner (see below).

Supplementary Figure 5: HeID- δ / ω competition and ϵ subunit.

a, Close-up view on δ^{NTD} and ω binding regions with ω transferred from an *E. coli* EC (PDB ID 6ALH) by superpositioning of the β subunits. δ^{NTD} and HeID displace the β' shelf and other secondary channel elements (black arrow), leading to steric hindrance of ω binding (red arrows).

On page 7 the authors describe unexplained density around HeIDBumper and model it as three consecutive helices of the DELTA-CTR. However, this reviewer thinks this is too speculative to be included in the coordinates for deposition because the provided maps are not of sufficient quality to support the model. The CTR portion should therefore not be deposited in my opinion.

We agree with the reviewer that we cannot reliably assign regions of δ to these map regions. However, we believe that the maps clearly reveal a ligand bound at the indicated sites. The only possibility for this ligand is δ CTR, which is consistent with our finding that HeID and δ form a binary complex dependent on the δ CTR and with δ /HeID cooperation in nucleic acid displacement. We stated in the original manuscript that our assignment of parts of the δ CTR is tentative. We have amended this statement in the revised manuscript to further underscore that no sequence can be fit to the density:

We observed some cryoEM density patches around HeID^{Bumper} that could only be interpreted as parts of δ ^{CTR} (Fig. 1a). However, the poor quality of the local cryoEM density did not permit reliable modeling of the precise region of δ ^{CTR} that bound at HeID^{Bumper}.

In addition, during the PDB submission we had already added a corresponding remark to the PDB headers concerning the part of the model in question:

OTHER_DETAILS: DUE TO LIMITED QUALITY OF THE ELECTRON DENSITY, WE WERE ONLY ABLE TO TRACE SOME OF THE PROTEIN MAIN CHAIN AS POLY-ALA

On page 8 the authors describe experiments that suggest a cooperative mode of action for HeID and the RNAP delta subunit in releasing DNA from RNAP (or preventing binding in the first place). It sounds like these experiments were done with DNA only (containing an artificial bubble) and all components were mixed at the same time. I think it would be very informative to do additional experiments and:

a) Pre-incubate RNAP alone with DNA prior to adding delta and/or HeID and compare to the present results

First, our sincere apologies – we have not properly presented the experimental setup for our δ /HeID/DNA competitive binding assays. In these assays, we indeed first formed RNAP-DNA complexes (formation of these complexes can be seen from the band shifts with RNAP in the absence of δ and HeID). δ and/or HeID were only added later. Thus, the assays indeed represent DNA displacement assays. We have corrected the description in the Methods section:

Nucleic acid displacement assays

Equimolar amounts of 5'-[³²P]-labeled ntDNA and unlabeled tDNA capable of forming an artificial bubble, or additionally an RNA 9-mer with complementarity to the tDNA in the bubble (Supplementary Table 1), were mixed in buffer G and annealed by heating to 95 °C for 5 min and subsequent cooling to 25 °C at 1 °C/min. 20 nM (to achieve a 10 nM concentration in the final mix) of the labeled DNA duplex or DNA/RNA scaffold were incubated with 2 μ M (to achieve a 1 μ M concentration in the final mix) RNAP ^{$\Delta\delta\Delta$ HeID} in buffer G for 10 min at 4 °C, followed by an additional 10 min incubation at 32 °C. Subsequently, (i) buffer, (ii) HeID (1 μ M final concentration); (iii) δ (1 μ M final concentration), (iv) combinations of HeID (1 μ M final concentration) and δ (titrated final concentration; see Fig. 3g) or (v) HeID and δ ^{NTD} (1 μ M final

concentration each) were added, and the samples were further incubated for 10 min at 32°C. Samples were loaded on a 4 % native PAGE gel and electrophoresed in 0.5X TBE buffer. Radioactive bands were visualized using a Storm phosphorimager and quantified using ImageQuant software (GE Healthcare).

b) Do the experiments also in presence of a complementary RNA transcript. This would potentially provide information if an EC is less susceptible to Delta/HeID action and if the order of events matter.

We thank the reviewer for this suggestion. For the original manuscript, we performed this assay with DNA harboring an artificial transcription bubble. We chose this DNA because it is expected to associate with RNAP in a manner that closely resembles a post-termination RNAP that remained stuck on the template (Ref. 40). Several recent papers show that RNAP frequently remains attached to DNA after termination (Ref. 3-5). Notably, such DNA-trapped RNAPs can initiate transcription in reverse direction (Ref. 5), which would pose a major threat to genome integrity due to head-on collisions with replisomes. Thus, we are convinced that our initial experiment monitors a highly relevant scenario of δ /HeID-mediated RNAP recycling. We now briefly present these arguments in the revised manuscript:

To further delineate the contributions of δ and HeID to nucleic acid displacement, we conducted band shift assays, in which we first bound RNAP to nucleic acids and subsequently added δ and/or HeID. We first tested displacement of DNA with an artificial bubble, which when bound to RNAP mimics a situation ensuing after many intrinsic termination events^{3-5,40}.

We fully agree with the reviewer that it is also important to monitor δ /HeID-mediated displacement of nucleic acids from an EC that contains DNA with a bubble stabilized by RNA, to assess whether the factors in principle can recycle stalled RNAPs. We therefore now also conducted these additional displacement assays. For analysis of EC disruption, we assembled ECs on a nucleic acid scaffold bearing an artificial bubble. As the DNA region forming the bubble cannot reanneal during displacement, EC disruption represents a particularly demanding task in our setup. We find that δ and HeID still quite efficiently displace the nucleic acids from such an EC, albeit less efficiently than DNA alone, as expected. We describe these additional results in the revised text and present them in revised Fig. 3g:

Next, we tested the ability of δ /HeID to dissociate ECs assembled on an artificial DNA bubble and complementary RNA, mimicking stalled ECs. A similar picture as for DNA-only displacement emerged; however, due to the RNA-mediated stabilization of DNA on RNAP, HeID and δ individually or HeID/ δ^{NTD} liberated less RNAP, and higher concentrations of δ in the presence of HeID were required to achieve full nucleic acid displacement (Fig. 3g, lanes 15-28).

Fig. 3: HeID/δ-mediated RNAP recycling.

g, EMSA monitoring displacement of DNA (lanes 1-14) or DNA/RNA (lanes 15-28) from RNAP by HeID, δ or combinations. Top scheme, samples analyzed; gray boxes, respective component added (proteins in equimolar amounts to RNAP^{ΔΔHeID}). Numbers, molar ratios of δ or δ^{NTD} relative to RNAP^{ΔΔHeID} added. Panels labeled “DNA” or “DNA/RNA”, native PAGE analyses. All lanes are from the same gel, some lanes for the DNA-only gel were removed for display purposes (dashed line). Bar graphs, quantification of the data shown in the middle panels. Values represent means of DNA bound relative to RNAP^{ΔΔHeID} alone ± SD for three independent experiments, using the same biochemical samples (data points indicated).

Also on page 8, the authors describe the effect of ATP (analogues) on HeID binding – does HeID hydrolyze ATP in presence (or absence) of RNAP? If the presence of RNAP causes ATP hydrolysis, it would explain why more HeID is released in presence of AMPPNP or ATP-gamma-S.

We thank the reviewer for pointing this out. HeID indeed exhibits intrinsic ATPase activity, which does not seem to be modulated by the presence of RNAP (Ref. 29). In addition, previous SAXS analyses had suggested conformational changes in HeID upon ATP binding (Ref. 27). We have amended our explanation accordingly:

Comparison of UvrD bound to DNA and ADP-Mg₂F₃³⁶ showed that the D1/D2 conformation of RNAP-bound HeID is incompatible with ATP binding (Fig. 6a). Since ATP binding to HeID induces conformational changes, as revealed by SAXS²⁷, we surmised that ATP-bound HeID may have a lower affinity for RNAP than the apo factor. Consistent with this notion, ATPγS, AMPPNP and, to a somewhat lesser extent, ATP led to the release of HeID from RNAP-δ-HeID during SEC, while ADP or AMP had minor effects (Fig. 6b; Supplementary Figure 7b). HeID exhibits intrinsic ATPase activity that is unaltered in the presence of RNAP²⁹. Thus, AMPPNP and ATPγS mimic conditions of constantly high ATP supply, whereas ATP is likely hydrolyzed and separated from RNAP/HeID during SEC, reducing its effect.

On page 9, the authors briefly describe an RNAP dimer - can they exclude that the presence of the detergent caused RNAP dimerization – negative stain (and to a lesser extent SEC) +/- detergent would provide a quick answer.

Again, we agree with the reviewer, and have conducted both experiments suggested: SEC-MALS and negative stain analyses without and with 0.15 % (w/v) n-octyl-glucoside with a sample otherwise identical to the sample used for cryoEM. As the critical micellar concentration of n-octylglucoside (0.6 % [w/v]) is well above the concentration we used for cryoEM (0.15 % [w/v]) there should be no interference from micelles. We indeed observe stabilization of dimers in the presence of n-octyl-glucoside but we also observe dimers in the absence of the detergent. We briefly describe these new findings in the revised text and have included additional figures for documentation (new Supplementary Figure 1d,e):

About two thirds of our cryoEM particle images conformed to dimeric (RNAP-δ-HeID)₂ complexes (Fig. 6c; Supplementary Note 3), which were partially stable during SEC under conditions identical to cryoEM sample preparation (0.15 % n-octylglucoside; Supplementary Figure 1d). We also conducted negative stain EM analyses with RNAP-δ-HeID in the presence or absence of 0.15 % n-octylglucoside and detected dimers under both conditions (Supplementary Figure 1e; a quantitative analysis of the monomer/dimer distribution was precluded by preferred particle orientations on the carbon films).

...

Intriguingly, we observed (RNAP-δ-HeID)₂ dimers resembling hibernating eukaryotic RNAP I (Fig. 6c,d), which were partially stable in SEC at initial RNAP concentrations about 10-fold lower compared to their nominal cellular concentrations in the log phase, estimated from transcript levels and ribosome profiling^{30,55}.

Supplementary Figure 1: Complex preparations.

d, SEC/multi-angle light scattering analysis of RNAP- δ -HeID used for cryoEM analysis in buffer lacking (-OG) or containing (+OG) 0.15 % (w/v) n-octylglucoside (critical micellar concentration 0.6 % [w/v]). Black traces, UV signals; red lines, molecular mass estimates across the peaks. Molecular masses deduced are listed in the bottom table compared to the theoretical (theor.) molecular masses for RNAP- δ -HeID (Mon.) and (RNAP- δ -HeID)₂ (Dimer). About 16 % of the sample traverses the column as intact (RNAP- δ -HeID)₂ dimers in the presence of n-octylglucoside.

e, Top, negative stain EM micrographs of RNAP- δ -HeID in buffer lacking (-OG) or containing (+OG) 0.15 % (w/v) n-octylglucoside. Scale bars, 100 nm. Bottom, 2D class averages of picked particle images. Classes boxed red unequivocally indicate the presence of (RNAP- δ -HeID)₂ dimers in both samples.

On page 22, the authors describe data collection and processing. They mention data was collected at 0.832Å/pixel, 4x binned for initial processing. However, at the bottom of page 22, they state particles were re-extracted at a pixel size of 1.3Å/pixel (consistent with the provided maps) – why were the images scaled to a different pixel size? This is not good practice as it may introduce interpolation artifacts and it is not necessary in this case because only one dataset was collected (so no need to combine data from different microscopes).

We agree with the reviewer that interpolation and binning represent a reduction in data completeness and precision. Initially, we refined our structures at a binning of 2 (pixel size 1.664 Å/px), which globally represents a sufficient sampling rate for the analyzed structure. Considering the local resolution of our reconstructions, we were limited by the pixel size and decided to use a finer sampling rate. The chosen pixel size of 1.3 Å/px was a good compromise between file size, achievable resolution and requirements of the hardware. Indeed, local resolution estimation was not limited any longer and map quality improved by visual inspection.

It appears the same box size was used for mono- and dimers. The dimers have dimensions of 260Å. A boxsize of 440 pixels (440*0.832a/pix = 280*1.307Å/pix = ~366Å) is very small for the dimer and may results in loss of high-resolution information, which is displaced as a result of the CTF (see Henderson and Rosenthal JMB 2003). The extent of displacement of high-resolution information depends on the applied defocus – e.g. at 2.5um defocus, the boxsize should be 1.5x bigger (i.e. ~620 pixels) to capture information to 4Å. The authors may consider a larger box size to gain more high-resolution information.

While we appreciate the insightful suggestion by the reviewer, we decided to keep the box size for the dimers comparatively small for several reasons. First, we wanted to be sure that only single, well separated particles are extracted. Secondly, the signal-to-background ratio is improved by using smaller boxes. As we used detergent for sample preparation, ice thickness is rather high reducing particle contrast and signal-to-noise ratio. Moreover, use of larger boxes will prohibit the use of consumer GPUs due to limited memory capacity and increase computational expense in general. Finally, using the same data acquisition setup and particle extraction box size (0.832 Å/px, 440 pixels) we have solved a ribosome complex of similar dimensions at 2.4 Å resolution (manuscript in review), suggesting that our resolution is not limited by the acquisition strategy or the refinement strategy.

Minor comments:

On page 4, the authors note that RNAP purified from the ΔHeID strain had essentially no omega subunit but RNAP purified from ΔHeID ΔDelta, did have omega – that seems odd and the reviewer wonders if the loss of omega is specific to the delta HeID strain or just an effect of differences in the purification stringency (i.e. omega is weakly bound and tends to dissociate).

We used the same purification strategy for all RNAP variants. Thus, it is unlikely that loss of ω in certain preparations is due to experimental differences. While ω is not essential and may be less stably integrated than other core RNAP subunits, the differential loss of ω from RNAP variants that contain δ was reproducible in our hands. We therefore believe that loss of ω is due to allosteric effects elicited by δ , which can be explained by comparing our structures with an *E.*

coli EC structure that includes ω . Please refer to our response to the second major point raised by the reviewer above.

Related to the previous comment, Extended Data Fig. 1C shows an SEC run of RNAP supplemented with HeID and Delta and the authors argue that omega was underrepresented – however, from the gel we cannot judge because omega likely ran out of the gel. Could the authors provide a gel, where we can see (substoichiometric) omega.

We now provide a new gel picture that includes the region to which ω runs, documenting our statement (new Supplementary Figure 1c):

The authors argue that the local resolution extends well beyond 3Å (Extended Data Fig. 3b). Judging from the maps, which were kindly provided, this reviewer thinks that it is likely overestimated and encourages the authors to use alternative software to independently confirm their estimates (e.g. blocres)

We have re-estimated local resolutions with blocres and obtained similar numbers. However, we have toned down our corresponding statement in the revised manuscript:

Refinement led to two maps for monomeric RNAP- δ -HeID and dimeric (RNAP- δ -HeID)₂ complexes at global resolutions of 4.2 Å and 3.9 Å, respectively, with local resolutions extending beyond these limits (Supplementary Figure 3b; Supplementary Table 2).

Reviewer #2 (Remarks to the Author):

The authors have satisfactorily addressed the issues raised during the reviews and I feel that the manuscript is stronger and more robust as a result.

Reviewer #3 (Remarks to the Author):

Thank you for submitting the revised version of this manuscript. The authors have addressed all my concerns and I recommend publication. I am looking forward to the final version.